# Observation of aligned dipoles and angular chromism of exciplexes in organic molecular heterostructures

Sang-hun Lee [1], Taek Joon Kim [1], Eunji Lee[2], Dayeong Kwon[1], Jeongyong Kim [2] ✉ & Jinsoo Joo [1] ✉

The dipole characteristics of Frenkel excitons and charge-transfer excitons between donor and acceptor molecules in organic heterostructures such as exciplexes are important in organic photonics and optoelectronics. For the bilayer of the organic donor 4,4′,4″-tris[(3-methylphenyl)phenylamino]triphenylamine and acceptor 2,4,6-tris(biphenyl-3-yl)-1,3,5-triazine molecules, the exciplexes form aligned dipoles perpendicular to the Frenkel excitons, as observed in back focal plane photoluminescence images. The angular chromism of exciplexes observed in the 100 meV range indicates possible delocalization and angle-sensing photonic applications. The blue shift of the peak position and increase in the linewidth of photoluminescene spectra with increasing excitation power are caused by the repulsive aligned exciplex dipole moments with a long lifetime (4.65 μs). Electroluminescence spectra of the exciplex from organic light-emitting diodes using the bilayer are blue-shifted with increasing bias, suggesting unidirectional alignment of the exciplex dipole moments. The observation of exciplex dipole moment alignments across molecular interfaces can facilitate the controlled coupling of exciton species and increase efficiency of organic light-emitting diodes.

Exciton species are fundamental quantum quasiparticles that are used in light-emitting and light-harvesting systems. Interlayer excitons (IXs), typically formed at heterojunctions (HJs) with type-II band alignment, exhibit distinctive properties compared with pristine excitons (e.g., Frenkel excitons (XFs) in organic semiconductors and Wannier excitons in inorganic semiconductors)[1–3]. Electrically or optically generated charge transfer (CT) induces the accumulation of electrons and holes at the lower conduction band minimum (CBM) and higher valence band maximum (VBM) across a HJ, respectively. CT exciton species, such as IXs, affect the quantum efficiency of optoelectronic and photonic devices[4,5]. For two-dimensional (2D) transition metal dichalcogenide (TMDC) systems, the spatially separated electrons and holes of IXs lead to a weak overlap between the wavefunctions of the electrons and holes. This results in a relatively long lifetime[1] that is applicable to future excitonic devices[2,6,7]. The spatial configuration of

IXs at a p-n (or donor-acceptor) HJ induces the unidirectional alignment of electric dipole moments perpendicular to the interface and the electric field dependence of the binding energy of IXs[1,8,9].

The CT excitons in heterostructures (HSs) can be generally formed in type-II band structure, which have been observed in not only TMDC-based HSs (e.g., MoS$_2$/WSe$_2$, MoSe$_2$/WSe$_2$, etc.) and WS$_2$/PbI$_2$[10] but also other various type-II band HSs such as TMDC/perovskite (WSe$_2$/(iso-BA)$_2$PbI$_4$) HS[11], perovskite/quantum dot (MAPbI$_3$/CdSe-ZnS-QD) HS[12], and PbS-CdS-QD/MAPbI$_{3-x}$Cl$_x$ HS[13]. Notably, the formation of CT excitons has been observed for the HSs of perovskites/QDs with rough interfaces[12,13]. The long lifetime and directional characteristics of electric dipole moments of the CT excitons were reported in the perovskite/QDs HSs[12,13].

For π-conjugated organic small/macro molecular systems, the wavefunction overlap of π-electrons and the π-π stacking of adjacent

[1]Department of Physics, Korea University, Seoul 02841, Republic of Korea. [2]Department of Energy Science, Sungkyunkwan University, Suwon 16419, Republic of Korea. ✉e-mail: j.kim@skku.edu; jjoo@korea.ac.kr

molecules enhance the delocalization of charges and intermolecular CT rate[14–16]. These characteristics of π-conjugated organic systems have resulted in the fabrication of advanced optoelectronic devices such as organic light-emitting diodes (OLEDs)[17–19], organic field-effect transistors[20], organic phototransistors[21], organic photovoltaic cells (OPVCs)[22,23], and organic waveguiding cavities[24,25]. Quantum quasi-particles similar to the IXs of TMDCs and perovskites/QDs HSs can be achieved in π-conjugated organic systems as exciplexes (XPs). In this study, the XPs represent the CT excitons formed in organic donor (D) and acceptor (A) heteromolecules[5,26,27]. These XPs have been extensively studied for OLEDs in terms of phosphorescence, thermally activated delayed fluorescence (TADF), and broadband emission such as in white-color-emitting OLEDs[19,28,29]. The XP binding energy ($E_B$) as a function of the wavefunction overlap and separation of electrons and holes in A and D molecules is an important factor for determining the efficiency of OLEDs[19,28–30] and OPVCs[31]. Another critical characteristic of XPs for applications in optoelectronic devices is the direction of the dipole moments in the D-A molecules and their correlation with the applied electric and local dipole fields. However, the directional features of the XF and XP dipole moments have not yet been directly evaluated in terms of luminance. Compared with 2D-TMDCs, CT excitons such as IXs and XPs in π-conjugated organic systems are formed at the molecular scale, and their exciton (XF) lifetimes (a few nanoseconds)[27,29,32] are an order of magnitude longer than those of TMDCs (a few hundred picoseconds)[33,34]. In addition, the lifetime of XPs in π-conjugated organic systems is longer than that of XFs[5,27,35]. Therefore, the long-lived and aligned XP is a promising energy host in optoelectronic devices or a platform for large interaction with other quasiparticles like photon and phonon in organic photonic devices.

4,4′,4′′-Tris[(3-methylphenyl)phenylamino]triphenylamine (m-MTDATA) and 2,4,6-tris(biphenyl-3-yl)-1,3,5-triazine (T2T) are the representative D and A organic molecules, respectively, for the study of exciton species. m-MTDATA has been widely used as the hole injection layer (HIL) in OLEDs because of its low ionization potential[36,37]. In previous studies on structure and crystallinity, m-MTDATA molecules were deposited on Au(111) or SiO2 substrates with in-plane columnar structures on the substrate surface[38,39], resulting in easy CT for XP generation. T2T has been widely used as the electron transport layer (ETL) in OLEDs owing to its electron-withdrawing triazine core and electron-rich biphenyl groups[18]. Naka-notani et al. reported long-range coupled XP with TADF based on a combination of m-MTDATA and T2T[27]. Because of its stable coupling, long lifetime, and high quantum efficiency, m-MTDATA/T2T is an ideal

HS for investigating the characteristics of XPs. The spatial long-range coupling of XPs in organic D-A HSs has been extensively studied in terms of the luminance characteristics depending on the thickness of the spacer layer including dopants[27] or the variation of D-A layers[28]. The directional features of the dipole moments of exciton species, including XPs, are important for optoelectronic applications such as OLEDs and OPVCs because the optical responsivity, energy transfer (ET) rate and angular dependency of light emission, and exciton dissociation efficiency can be determined based on the orientation of the exciton dipole moments of the exciton species. However, the dipole directional characteristics of XFs and XPs and their correlations with organic HSs have not yet been systematically studied.

Here, the distinctive orientations of the dipole moments of the exciton species in m-MTDATA, T2T, and m-MTDATA/T2T bilayer HSs were directly demonstrated by mapping the back focal plane (BFP) photoluminescence (PL) spectra. The electric dipole moments corresponding to the XPs from the m-MTDATA/T2T bilayer ($XP_{m-MT/T2T}$) HS were orthogonal to those of the XFs of the m-MTDATA monomers ($XF_{m-MT}$) (or to the interface of the HJ). The observed angular dispersion of the emission energy of $XP_{m-MT/T2T}$ is the interesting demonstration of the angular chromism of XPs in π-conjugated organic molecular HSs. With increasing incident excitation power ($P_{in}$), the PL peaks of XPs were blue-shifted, and the full width at half maximum (FWHM) of XPs increased. This indicated the enhancement of the repulsive dipole alignment between XPs. The lifetime of the delayed fluorescence of $XP_{m-MT/T2T}$ measured via time-resolved PL was considerably long (2.14 μs at 290 K), suggesting the occurrence of reverse intersystem crossing (RISC), spatial separation of electron and hole for $XP_{m-MT/T2T}$, and stronger repulsive dipole interactions between XPs. The PL peak positions of the XPs were blue-shifted, whereas those of the XFs remained constant with decreasing temperature. This confirmed the anisotropy of the dipole directions of XFs and XPs in the m-MTDATA/T2T D-A bilayer. The blue shift of the electroluminescence (EL) peaks corresponding to the XP with increasing forward bias observed for the m-MTDATA/T2T bilayer organic light-emitting diode (OLED) supported the alignment of dipole moments.

## Results
### PL and energy-band structure
Figure 1a, b show the schematic chemical structures of the m-MTDATA donor and T2T acceptor molecules, respectively. For the preparation of high-quality samples of m-MTDATA single layer, T2T single layer, and m-MTDATA/T2T bilayer thin films, the corresponding molecules

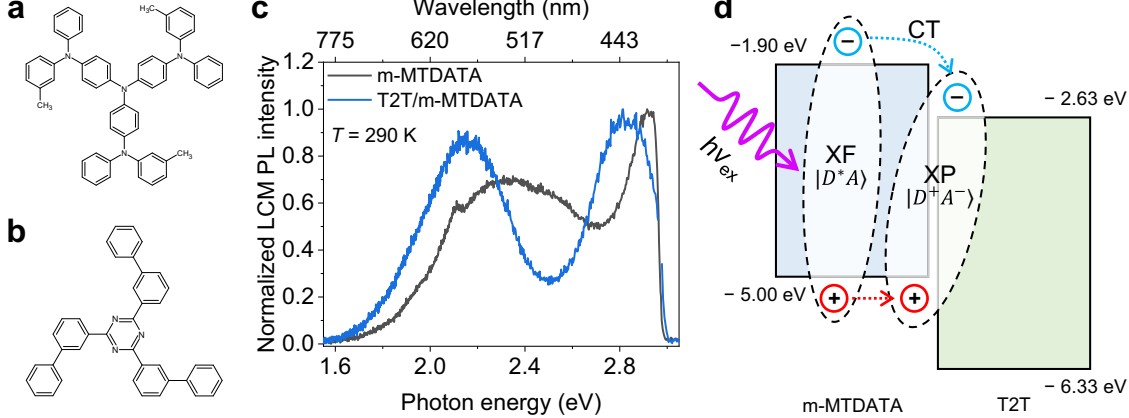

**Fig. 1 | Photoluminescence spectra and band alignment of donor and acceptor molecules.** Schematic chemical structures of **a** m-MTDATA and **b** T2T molecules. **c** Normalized LCM PL spectra of m-MTDATA (black curve) and m-MTDATA/T2T bilayer (blue curve) thin films at 290 K. The PL peak due to the XP was observed at 2.16 eV. **d** Energy-band alignment with HOMO and LUMO levels of m-MTDATA and T2T films and the formation of photoinduced CT exciton between m-MTDATA donor and T2T acceptor (i.e., XP). The HOMO and LUMO levels were estimated by using UPS and optical absorption spectra.

were deposited on a Si/SiO₂ substrate at an extremely low rate of deposition (0.3 nm min⁻¹) using organic molecular beam deposition (OMBD) system. This system had two crucibles with holes of 1.5 mm diameter for beam injection of organic molecules. The high-vacuum environment (under $5.0 \times 10^{-6}$ Torr) during deposition reduced the defects, and the fine control of the flux rate of the molecular beam contributed to the epitaxial growth of high-quality organic layers. The normalized laser confocal microscopy (LCM) PL spectra of the m-MTDATA (black curve) and m-MTDATA/T2T (blue curve) films are shown in Fig. 1c. The spectra were recorded at room temperature (290 K) in a high-vacuum cryostat chamber using a 405-nm diode laser with $P_{in}$ = 0.1 μW. The XF peak of m-MTDATA monomer ($XF_{m-MT}$) was observed at 2.90 eV (~427 nm). The broad PL peak observed at 530 nm was attributed to the excimer ($XM_{m-MT}$) emission peak from m-MTDATA. The PL peak at approximately 530 nm of $XM_{m-MT}$ broadened with increasing temperature and exhibited a longer decay time than that of $XF_{m-MT}$, as depicted in Supplementary Fig. 1. These results are consistent with the typical characteristics of perylene XMs[36,40]. The LCM PL spectrum of the m-MTDATA/T2T bilayer (blue curve in Fig. 1c) exhibited the PL emission of the XP in the bilayer ($XP_{m-MT/T2T}$) at 2.16 eV (~575 nm), wherein the $XM_{m-MT}$ was partially included. It is noted that the observed $XP_{m-MT/T2T}$ is due to the CT between m-MTDATA donor and T2T acceptor. Supplementary Fig. 2 shows the raw data (i.e., absolute intensity) of LCM PL spectra ($\lambda_{ex}$ = 405 nm and $P_{in}$ = 0.1 μW) of the m-MTDATA and T2T layers. The PL intensity of the T2T layer was weaker than that of the m-MTDATA layer because of the low absorbance of T2T at $\lambda_{ex}$ = 405 nm, suggesting the exclusion of the T2T component from the deconvoluted LCM PL spectra.

The HOMO and LUMO levels and energy-band alignments (EBAs) of the m-MTDATA/T2T bilayer are shown in Fig. 1d; these were determined from ultraviolet photoelectron spectroscopy (UPS) and UV/visible absorption spectra (Supplementary Fig. 3). The HOMO levels of m-MTDATA and T2T were −5.00 and −6.33 eV, respectively. The LUMO levels of m-MTDATA and T2T were −1.90 and −2.63 eV, respectively. The details are presented in Supplementary Fig. 3. The EBA of our m-MTDATA/T2T bilayer exhibited a type-II band alignment, resulting in a photoinduced CT and the formation of XP at the interface, as shown in Fig. 1d. The quantum states of a photoexcited D-A molecular system ($[DA]^*$) can generally be described as $|[DA]^*\rangle = a_1|D^*A\rangle + a_2|DA^*\rangle + a_3|D^+A^-\rangle + a_4|D^-A^+\rangle$, where $|[DA]^*\rangle$, $|D^*A\rangle$, $|DA^*\rangle$, $|D^+A^-\rangle$, and $|D^-A^+\rangle$ are the wavefunctions of the $[DA]^*$, D-excited, A-excited, and electron-transferred terms from D to A and A to D, respectively. The type-II band alignment between the T2T and m-MTDATA molecules, as shown in Fig. 1d, reveals the dominant transfer of electrons from m-MTDATA to T2T, while being $a_4$ negligible. When the excited electrons in m-MTDATA are dominant (Supplementary Fig. 2), $a_2$ becomes negligible. Therefore, the final quantum state of $[DA]^*$ can be described as $|[DA]^*\rangle = a_1|D^*A\rangle + a_3|D^+A^-\rangle$, as shown in Fig. 1d. In this study, $a_3|D^+A^-\rangle$ represents the quantum state of XP between m-MTDATA donor and T2T acceptor. The measured PL peak corresponding to XP is 575 nm (=2.16 eV), as indicated in Fig. 1c, which is smaller than the directly estimated value (=2.37 eV) at approximately 210 meV. This is because the high binding energy ($E_B$) of the HOMO levels of organic molecules measured by performing UPS is downshifted owing to the polarization energy induced by photo-induced holes[40–42].

## BFP PL

Figure 2a shows a schematic of the BFP PL experiment. The lateral component of the wavevector ($k_x$) of the photons emitted from the sample can be described as $k_x = k_0\sin\theta$, where $k_0$ and $\theta$ denote the magnitude of the wavevector in air and emission angle of the emitted photon, respectively. The maximum $\theta$ ($\theta_{max}$) is calculated as, $\theta_{max} = \sin^{-1}(NA/n)$, where NA and $n$ are the numerical aperture of the objective lens and refractive index of the medium, respectively. The

BFP PL images of the samples were obtained by using a charged coupled device (CCD) camera at 290 K ($\lambda_{ex}$ = 405 nm) employing appropriate optical filters. Figure 2b, c present the BFP PL images of the m-MTDATA/T2T bilayer corresponding to $XF_{m-MT}$ (405 nm < $\lambda$ < 450 nm) and $XP_{m-MT/T2T}$ (488 nm < $\lambda$ < 600 nm), respectively. The white curves in Fig. 2b, c are proportional to the BFP PL intensity, as indicated by the yellow dotted lines. From the images, the center part in Fig. 2b and the edge part in Fig. 2c are brighter than other parts in the corresponding spectral range. In BFP experiments, the more intense emission is expected at the center of BFP than toward the edge of BFP owing to low angle emission from the horizontal (in-plane) orientation of dipoles for XFs. With a similar analogy, a relatively more intense emission can be detected toward the edge of the BFP than at the center of the BFP owing to the high-angle emission from the vertical (out-of-plane) orientation of dipoles of XPs. The convexity of $XF_{m-MT}$ and concavity of $XP_{m-MT/T2T}$ in the white curves are evident, demonstrating the distinct directions of the dipole moments of $XF_{m-MT}$ (i.e., intra-molecular excitons) and $XP_{m-MT/T2T}$ (hetero-intermolecular excitons) in the m-MTDATA/T2T bilayer. As the orientation of the XF dipoles is relatively random in the lateral plane, and as XP possesses aligned dipoles across the interfaces between the m-MTDATA and T2T layers, the emission angles ($\theta_{em}$) of $XF_{m-MT}$ and $XP_{m-MT/T2T}$ must be mutually orthogonal. Therefore, BFP PL mapping directly reveals the orthogonality of the XF and XP of the m-MTDATA/T2T bilayer.

To confirm the BFP CCD images, the BFP PL spectra were mapped. Figure 2d displays the BFP PL spectra of the m-MTDATA/T2T bilayer at 3 K. The BFP PL spectra were averaged over the center (black curve) and edge (red curve) regions of the BFP mapping images. The BFP PL spectrum of XP (including XM) was blue-shifted from 2.089 eV (center) to 2.194 eV (edge), indicating the dispersion of the XP energy. The protruding PL peak in region I ($\lambda$ = 420–430 nm) corresponds to the 0-0 (non-vibrational) mode peak of $XF_{m-MT}$. The broad and intense PL peak in region II ($\lambda$ = 520–650 nm) corresponds to XP including XM[43]. The BFP PL mapping images of the m-MTDATA/T2T bilayer confirm the distinctive anisotropy between the $XF_{m-MT}$ and $XP_{m-MT/T2T}$ (including XM) dipole moments, as shown in Supplementary Fig. 4a, b, respectively. Figure 2e shows the normalized PL intensity of $XF_{m-MT}$ (black) and $XP_{m-MT/T2T}$ (red) as a function of $k_x/k_0$ from the crossline A-B of the BFP presented in Supplementary Fig. 4a, b; $k_x$ represents the component of the in-plane wavevector of the emitted photon ($k_x = k_0\sin\theta$). The concavity of $XP_{m-MT/T2T}$ and convexity of $XF_{m-MT}$ as a function of $k_x/k_0$ provide a visual indication of the orthogonality of the exciton dipole moments; that is, $XP_{m-MT/T2T}$ was out of plane to that of $XF_{m-MT}$. Similar IX behaviors were observed in TMDC-based HSs[44–46]. In previous literature, the edge of BFP near at $k_x = \pm 1$ often displays intense PL regardless of the exciton orientation[44,46], due to the highly efficient outcoupling near at supercritical angle[47]. However, the NA of the air objective lens used in our BFP PL imaging limits the range of $k_x$ well within $\pm 1$ and thus the scheme shown in Fig. 2a is generally valid, showing the characteristic convexity and concavity in BFP profile for in-plane or out-of-plane excitons, respectively[46].

The k-dependent PL spectra measured from the BFP, that is, the energy-momentum (E-k) features of the m-MTDATA/T2T bilayer, are shown in Fig. 2f. The broad and intense PL of $XP_{m-MT/T2T}$ at 2.098–2.209 eV (red-orange color) was observed in the full range of $k_x/k_0$, whereas that of $XF_{m-MT}$ at 2.90 eV (=427.6 nm; green color region between white dotted lines) was observed at a small emission angle (|$k_x/k_0$| ≤ 0.3). Overall, the higher PL intensity was observed at large emission angle (|$k_x/k_0$| ≥ 0.4) because of the dominance of $XP_{m-MT/T2T}$ in PL of our HS through efficient CT across the m-MTDATA/T2T interface. Moreover, the emission energy of $XP_{m-MT/T2T}$ exhibited an obvious angular dispersion, where the emission peak position of $XP_{m-MT/T2T}$ was blue-shifted as |$k_x/k_0$| increased, implying the energy-momentum dispersion characteristics. We emphasize that the angular dispersion of the emission energy of $XP_{m-MT/T2T}$ is the demonstration

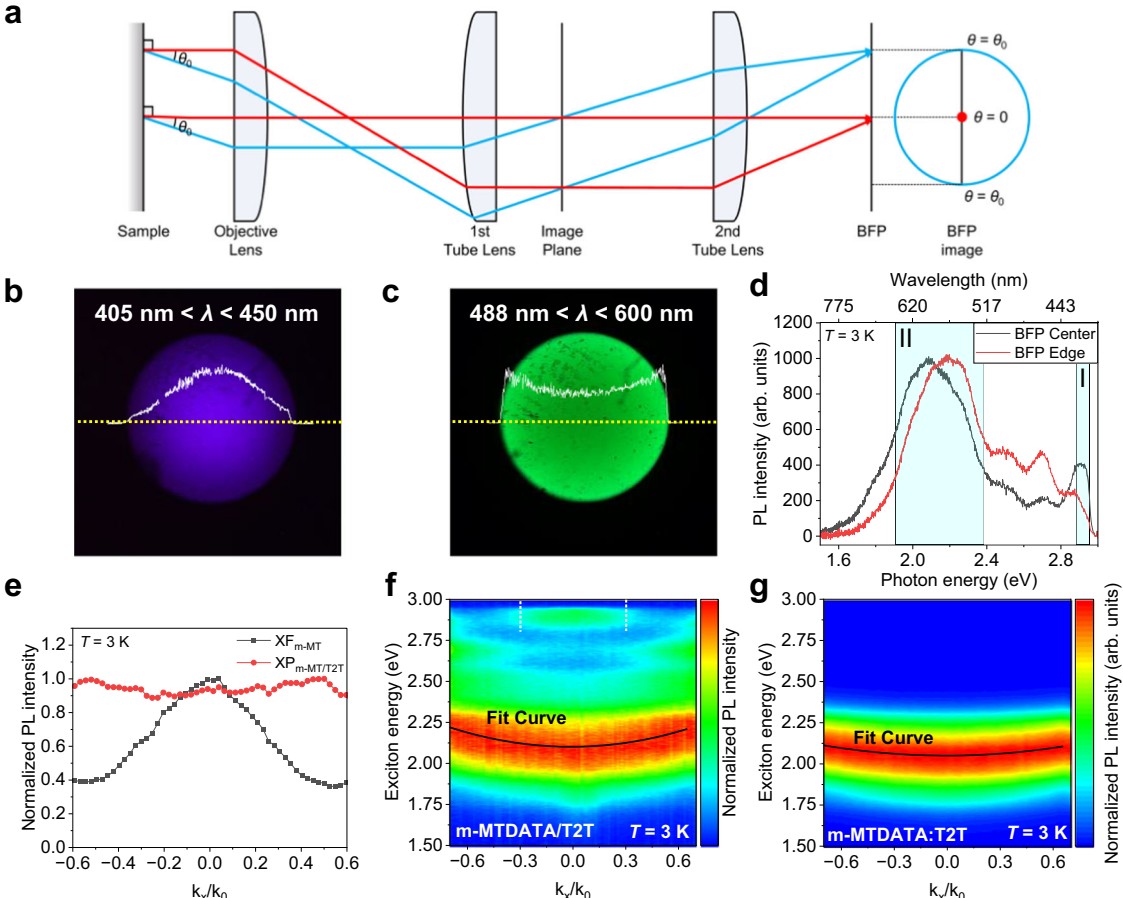

**Fig. 2 | Back focal plane photoluminescence. a** Schematic illustration of BFP PL measurement. BFP images of m-MTDATA/T2T bilayer captured by CCD camera at 290 K ($\lambda_{ex}$ = 405 nm). The detection ranges of wavelength are **b** 405 nm < $\lambda$ < 450 nm ($XF_{m\text{-}MT}$) and **c** 488 nm < $\lambda$ < 600 nm ($XP_{m\text{-}MT/T2T}$). **d** Averaged BFP PL spectra of m-MTDATA/T2T bilayer film. The spectra were averaged in the center (black curve) and edge (red curve) regions of BFP PL mapping. **e** Normalized PL intensity of $XF_{m\text{-}MT}$ (black) and $XP_{m\text{-}MT/T2T}$ (red) as a function of $k_x/k_0$. Normalized $k$-dependent BFP PL spectra (exciton energy) of **f** m-MTDATA/T2T bilayer and **g** m-MTDATA:T2T CDL with parabolic fitting curves (black) at 3 K.

of angular chromism of XP in organic molecular HSs. Angular chromism can occur in crystals with molecular lattice anisotropy. No angular dispersion was observed for $XF_{m\text{-}MT}$ because of the strongly localized nature of $XF_{m\text{-}MT}$ within the m-MTDATA molecules. Because XPs are the result of the CT between D-A heteromolecules or interlayers, their delocalized nature and long-range interactions are attributed to the angular chromism of $XP_{m\text{-}MT/T2T}$. The delocalization here represents relatively broad overlap of wavefunctions between aligned dipoles of XPs in the bilayer due to the large spatial extension of XP wavefunction. The distinct observation of the angular chromism of XP in our HS implies promising photonic applications in angle-dependent sensing, displays, filters, switches, or information encoding and decoding algorithms.

The m-MTDATA and T2T co-deposition layers (CDLs) were fabricated, and their BFP PL mapping images were compared with those of the bilayer films. The m-MTDATA and T2T molecules were simultaneously co-deposited with the same concentration ratio (1:1) on the Si/SiO$_2$ substrate (at 0.3 nm min$^{-1}$) in the OMBD chamber. With the normalized PL intensity as a function of $k_x/k_0$ (Supplementary Fig. 5b), the convexity of $XF_{m\text{-}MT}$ and concavity of the exciplex in the CDL ($XP_{m\text{-}MT:T2T}$) were still observed. The emission energy ($E$) as a function of $k_x/k_0$, that is, the energy-momentum ($E$-$k$) dispersion relations of $XP_{m\text{-}MT:T2T}$ in the CDL, at 3 K are shown in Fig. 2g. Notably, the PL intensity of $XF_{m\text{-}MT}$ was extremely weak in the CDL because of the active CT from m-MTDATA to T2T and the prevailing formation of XPs over Frenkel excitons. The PL spectra were deconvoluted (Supplementary

Fig. 6) and the PL peak position of XP between m-MTDATA and T2T was fitted by applying the parabolic function of $E(k) = \hbar^2 k^2 / 2m^* + E_{XP(k=0)}$ (Fig. 2f, g); $h$ (= $2\pi\hbar$) is the Planck constant and $m^*$ is the effective mass. To quantify the degree of angular dispersion of XP PL energy, we define $\kappa_{center} \equiv d^2E/dk^2$, that is, $m^* = \hbar^2/(d^2E/dk^2) = \hbar^2/\kappa_{center}$. The $\kappa_{center}$ of the XP peak of the m-MTDATA/T2T bilayer was estimated to be 0.499 at 3 K, which was larger than that (= 0.259) of m-MTDA-TA:T2T CDL. This is consistent with the more delocalized nature of $XP_{m\text{-}MT/T2T}$ in the bilayer based on CT across the bilayer interface than that of $XP_{m\text{-}MT:T2T}$ in the CDL based on local CT between mixed m-MTDATA and T2T molecules. Further, the directions of the XP dipole moments between m-MTDATA and T2T were randomly up and down (Supplementary Fig. 5a) owing to the co-deposition of the D and A molecules. Our results also indicated the easy diffusion of $XP_{m\text{-}MT/T2T}$ in the bilayer and its potential use in long-range exciplex transport.

To understand the relationship between the XP $E$-$k$ dispersion and the degree of dipole alignment of the XPs, additional experiments using more disordered films of donor m-MTDATA and acceptor T2T blending, where the orientation of XPs should be all random, were performed. The $E$ vs. $k$ relations of the XP peak from the BFP PL spectra and images of the blended drop-cast and reprecipitated films did not show the dispersive characteristics as shown in Supplementary Figs. 7 and 8. The negligibly small $\kappa_{center}$ of drop-cast and reprecipitated films compared to those of bilayer and CDL by approximately an order of magnitude are listed in Supplementary Table 1. In addition, while XPs of drop-cast or reprecipitated films displayed the clear XP emission at

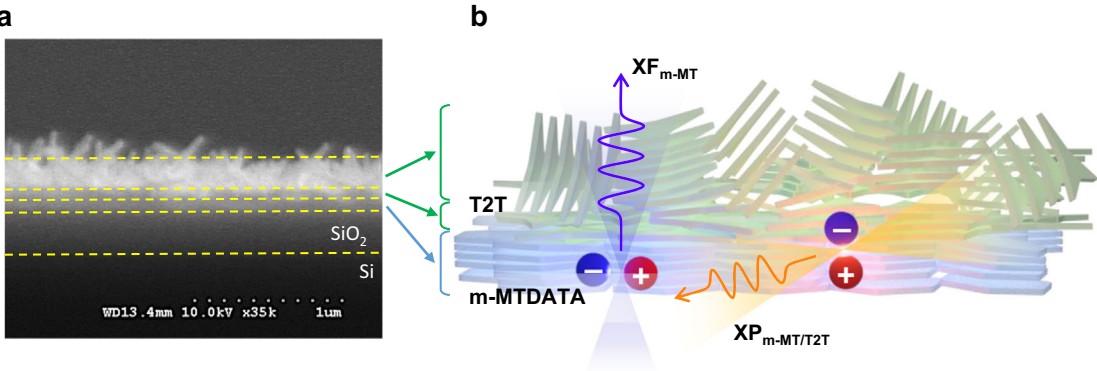

**Fig. 3 | Stacking structure of m-MTDATA/T2T bilayer. a** Cross-sectional SEM image of m-MTDATA/T2T bilayer. **b** Schematic illustration of the molecular stacking and the formation of dipoles and distinctive emission directions of intramolecular XF in m-MTDATA ($XF_{m\text{-}MT}$; blue) and hetero-intermolecular charge transfer excitons ($XP_{m\text{-}MT/T2T}$; orange) in m-MTDATA/T2T bilayer.

the same wavelength as BL samples, the characteristic concavity of BFP cross-sectional profile was not observed in either of additionally prepared samples (Supplementary Fig. 7), confirming that observed concavity of BFP profile in BL samples originates from the vertically aligned nature of XP dipole orientation.

The m-MTDATA and T2T are π-conjugated molecules exhibiting a planar shape. The deposition of m-MTDATA molecules on the Si/SiO₂ substrate below the glass transition temperature exhibits an ordered in-plane columnar structure[39]. Figure 3a shows the cross-sectional SEM image of m-MTDATA/T2T bilayer. To investigate molecular orientations, the indices of refraction of m-MTDATA, T2T, and their bilayer are estimated using ellipsometer experiments as shown in Supplementary Fig. 9. The ordinary ($n_o$) and extraordinary ($n_e$) refraction indices of the m-MTDATA layer were calculated to be 1.72 and 1.70 (negative birefringence) at $\lambda = 550$ nm, agreeing with previous results[48]. The calculations of ellipsometer experiments of T2T layer and m-MTDATA/T2T bilayer are more complicated (refer to Supplementary Fig. 9b). The molecular orientations and layer structure of m-MTDATA/T2T bilayer based on the ellipsometer analysis combined with the cross-sectional SEM and TEM images (Supplementary Figs. 9–11) are suggested that m-MTDATA and T2T molecules were in-plane stacking at the interface and then the T2T molecules were randomly stacked out-of-plane at the top layer as the form of nanorods. Figure 3b shows the systematic illustration of molecular stacking and the formation of dipoles and distinctive emission from intramolecular XFs (blue) in m-MTDATA and hetero-intermolecular CT excitons (XPs; orange) in the m-MTDATA/T2T bilayer. The dipole directions of the intramolecular $XF_{m\text{-}MT}$ and hetero-intermolecular CT excitons (exciplexes; $XP_{m\text{-}MT/T2T}$) in the m-MTDATA/T2T bilayer were perpendicular to each other.

## Power dependency of PL at low temperatures

Figure 4a shows the normalized LCM PL spectra at 3 K at various excitation laser powers ($P_{in}$). The PL peaks for $XF_{m\text{-}MT}$ and $XP_{m\text{-}MT/T2T}$ in the bilayer were observed at 2.916 eV (=425.2 nm) and 2.147 eV (=577.6 nm), respectively, at 3 K and $P_{in} = 10$ nW. The PL spectra with $XF_{m\text{-}MT}$ and $XP_{m\text{-}MT/T2T}$ peaks in the bilayer became clearer with increasing $P_{in}$. The $P_{in}$-dependent LCM PL peak position of the m-MTDATA/T2T bilayer was measured at various low temperatures, as shown in Fig. 4b, c and Supplementary Fig. 12. The PL peak positions of $XP_{m\text{-}MT/T2T}$ (red markers) were weakly blue-shifted from 2.145 to 2.160 eV at 3 K and from 2.143 to 2.154 eV at 50 K as $P_{in}$ increased from 4.0 nW to 5.0 μW. However, the PL peak positions of $XF_{m\text{-}MT}$ were relatively constant (2.913–2.914 eV at all measured temperatures) with increasing $P_{in}$, as shown in Fig. 4b, c and Supplementary Fig. 12. The FWHM of PL spectra of $XP_{m\text{-}MT/T2T}$ in the bilayer increased from 0.42 to 0.47 eV at 3 K, whereas that of $XF_{m\text{-}MT}$ was constant at approximately

0.098 eV, with an increase in $P_{in}$ from 20.0 nW to 5.0 μW, as shown in Fig. 4d and Supplementary Fig. 13. Therefore, the PL peak positions and FWHM variations of $XP_{m\text{-}MT/T2T}$ as a function of $P_{in}$ were different from those of $XF_{m\text{-}MT}$. Similar behaviors of the peak positions and FWHM of the PL spectra were observed at other low temperatures (100, 150, 200, 250, and 290 K), as shown in Supplementary Figs. 12 and 13. The blue shift of the peak position and increase in the FWHM of the PL spectra of $XP_{m\text{-}MT/T2T}$ originated from the repulsive dipole-dipole interaction caused by the increase in the $XP_{m\text{-}MT/T2T}$ concentration with increasing $P_{in}$[9–12,49]. The blue shift of the XP peak with increasing $P_{in}$ is similar to that of the IXs for TMDC-based HSs[1,49,50]. With increasing power of excitation laser, the concentration of accumulated charges at the heterojunction of D-A interface increased, resulting in the increase of density of dipoles corresponding to XPs and enhancing the repulsive dipole-dipole interaction. The similar results have been observed in other HSs such as WSe₂/(iso-BA)₂PbI₄[11] and MAPbI₃/CdSe-ZnS-QD HS[12] with their rough interfaces. The relationship between PL blue-shift and repulsive dipole interaction had been theoretically and experimentally studied in indirect excitons of coupled quantum wells[51,52].

The relationship between the PL intensity ($I_{PL}$) and $P_{in}$ ($I_{PL} = I_0 P_{in}{}^\alpha$) was investigated to understand the exciton recombination processes of the PL emission at 3 K, as shown in Fig. 4e. The $\alpha$ values were estimated to be 0.81 and 0.72 for $XF_{m\text{-}MT}$ and $XP_{m\text{-}MT/T2T}$, respectively. An $\alpha$ value close to unity is expected for free exciton transition caused by single photon absorption, whereas a value below 1.0 is expected for easy saturation of excited states[1,3,11]. Defect states[53], D (A) states[54], and IX excitons with long lifetime[i] induce easy saturation. As temperature increased to 290 K, the values of $\alpha$ of the $XP_{m\text{-}MT/T2T}$ increased from 0.73 (3 K) to 0.90 (290 K), as shown in Fig. 4f, similar to the values of (0.6–0.9) for IXs in 2D-TMDCs[1,3,10]. The $XP_{m\text{-}MT/T2T}$ are loosely bound to molecular heterojunctions compared with $XF_{m\text{-}MT}$, resulting in lower $\alpha$ values in the equation of $I_{PL} = I_0 P_{in}{}^\alpha$.

## Lifetimes of exciton species and temperature dependency of PL

The time-resolved PL (tr-PL) decay characteristics were investigated to determine the lifetimes of the XF of the m-MTDATA layer and XP of the m-MTDATA/T2T bilayer. The tr-PL decay curves were fitted with a double-exponential decay curve. The insets in Fig. 5a–c display the tr-PL decay curves of $XF_{m\text{-}MT}$, prompt fluorescence of $XP_{m\text{-}MT/T2T}$, and delayed fluorescence of $XP_{m\text{-}MT/T2T}$, respectively, at various low temperatures. The average lifetimes ($\tau_{avg}$) of $XF_{m\text{-}MT}$ and $XP_{m\text{-}MT/T2T}$ as a function of temperature are estimated by using the intensity weighted average lifetime, $\tau_{avg} = \Sigma(a_i\tau_i{}^2)/\Sigma(a_i\tau_i)$ ($i = 1, 2$), as shown in Fig. 5a–c. The $\tau_{avg}$ of the XF for the m-MTDATA layer at 3 K is estimated to be 1.05 ns. As shown in Fig. 5a, $\tau_{avg}$ of $XF_{m\text{-}MT}$ increases from 1.05 to 1.22 ns as

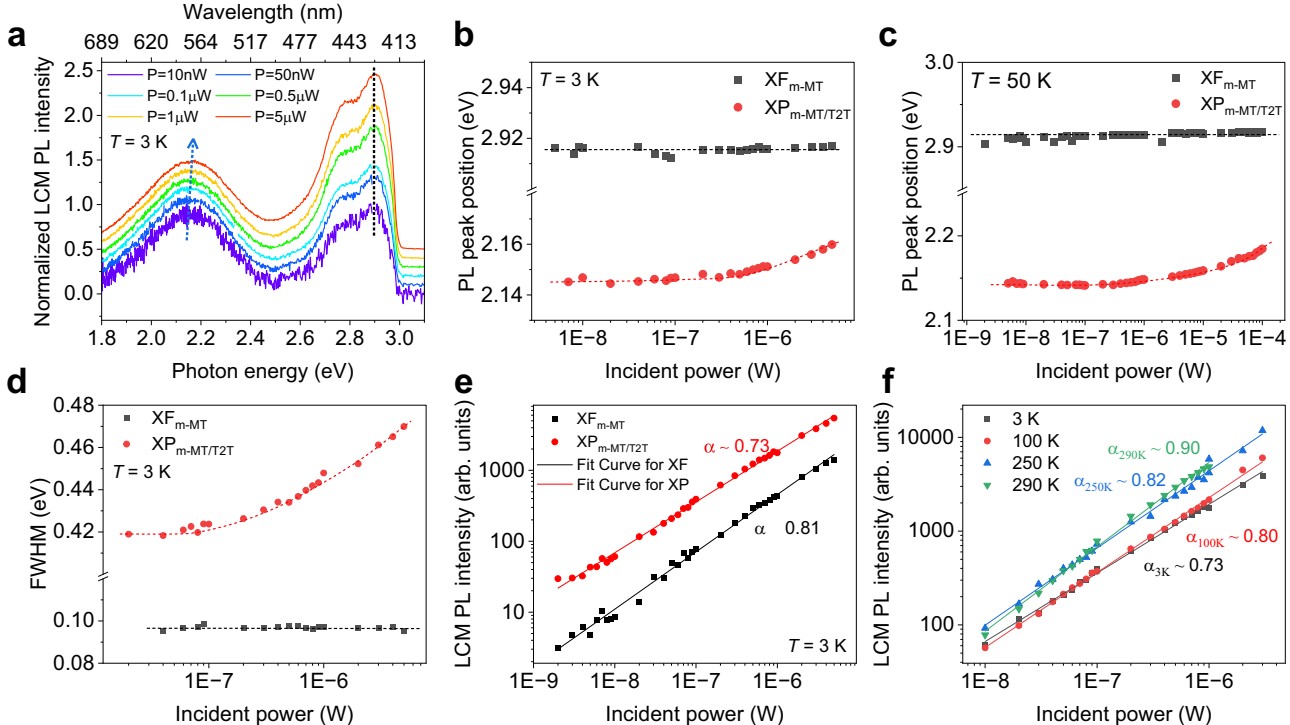

**Fig. 4 | Incident power dependence of LCM PL. a** Normalized LCM PL spectra of m-MTDATA/T2T bilayer at 3 K with various excitation powers ($P_{in}$). LCM PL peak position of XF$_{m-MT}$ (black markers) and XP$_{m-MT/T2T}$ (red markers) at **b** 3 K and **c** 50 K. **d** FWHM of LCM PL spectra of XF$_{m-MT}$ (black markers) and XP$_{m-MT/T2T}$ (red markers) as a function of $P_{in}$ at 3 K. LCM PL intensity as a function of $P_{in}$ in logarithmic scale for **e** XF$_{m-MT}$ and XP$_{m-MT/T2T}$ at 3 K, and **f** XP$_{m-MT/T2T}$ at various low temperatures.

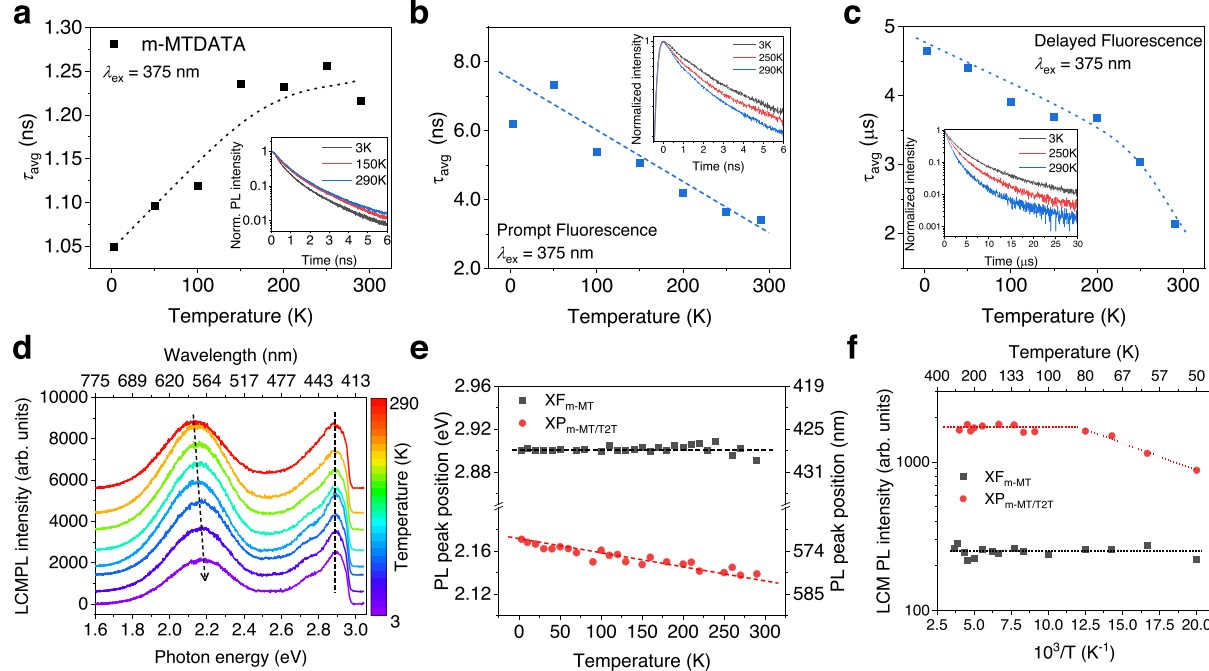

**Fig. 5 | Temperature dependence of time-resolved PL and LCM PL.** Average lifetimes ($\tau_{avg}$) of **a** XF$_{m-MT}$, **b** prompt fluorescence of XP$_{m-MT/T2T}$, and **c** delayed fluorescence of XP$_{m-MT/T2T}$, as a function of temperature. Inset: tr-PL decay curves of the corresponding exciton species at various low temperatures (3 K, 250 K, and 290 K). **d** LCM PL spectra of m-MTDATA/T2T bilayer at various low temperatures ($\lambda_{ex}$ = 405 nm and $P_{in}$ = 0.1 μW). Temperature-dependent **e** PL peak positions and **f** LCM PL intensity obtained by deconvoluting the corresponding LCM PL spectra of m-MTDATA/T2T bilayer. The black and red makers indicate the XF$_{m-MT}$ and XP$_{m-MT/T2T}$ components of the bilayer, respectively.

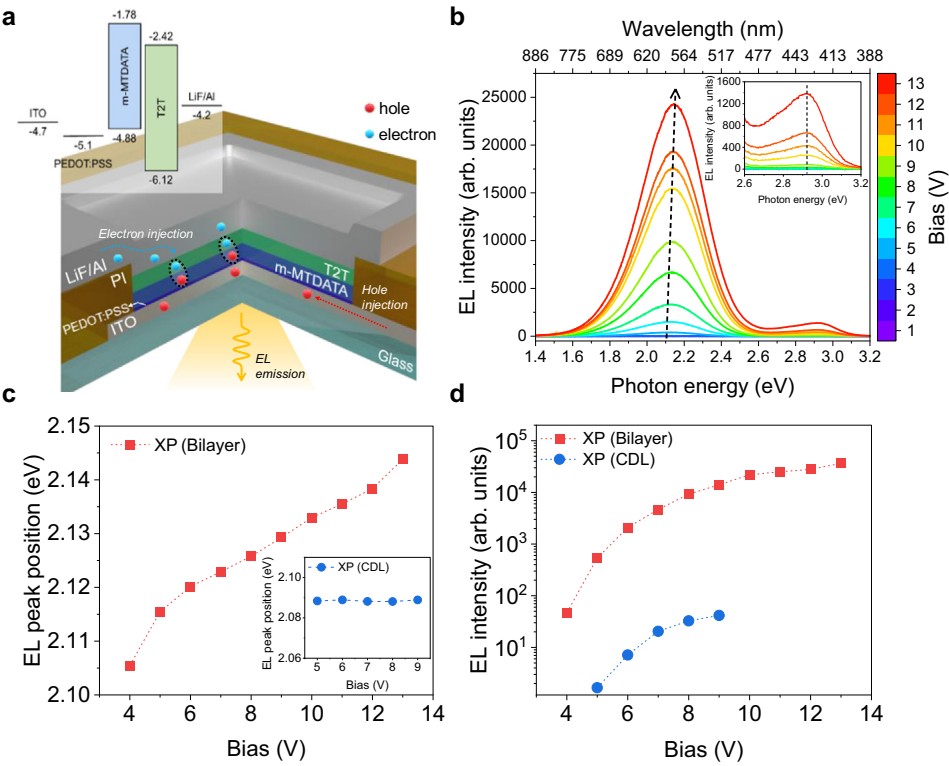

**Fig. 6 | Electroluminescence of BL and CDL OLEDs using m-MTDATA and T2T.**
**a** Schematic device structure and energy-band diagram of the OLED using m-MTDATA/T2T bilayer. **b** EL spectra of m-MTDATA/T2T bilayer OLED in various applied biases. Inset: Magnification of Fig. 5b in the range of 2.6–3.2 eV for $XF_{m\text{-}MT}$.

**c** EL peak positions for the $XP_{m\text{-}MT/T2T}$ bilayer OLED (red markers) as a function of applied bias. Inset: EL peak positions (blue markers) for the $XP_{m\text{-}MT:T2T}$ CDL OLED. **d** EL intensities of $XP_{m\text{-}MT/T2T}$ in the bilayer (red markers) and CDL (blue markers) OLEDs as a function of applied bias.

temperature increases from 3 to 290 K. The increase in the $\tau_{avg}$ of m-MTDATA with increasing temperature is attributed to the high quantum yield of m-MTDATA[29]. The $\tau_{avg}$ of the T2T layer is shown in Supplementary Fig. 14. Two components for the long $\tau_{avg}$ of $XP_{m\text{-}MT/T2T}$ correspond to the prompt (3.43–7.33 ns) and delayed (2.14–4.65 μs) fluorescent decay by TADF[27], as shown in Fig. 5b, c, respectively. The values of $\tau_{avg}$ of the prompt and delayed fluorescence decreased as temperature increased from 3 to 290 K owing to the activation of non-radiative decay and the activation of reverse intersystem crossing (RISC). This is consistent with the previous reports[5,55]. Notably, the measured extremely long $\tau_{avg}$ of the m-MTDATA/T2T bilayer (TADF) were varied from 4.65 μs (3 K) to 2.14 μs (290 K). The long lifetime of $XP_{m\text{-}MT/T2T}$ from the cryogenic temperature to room temperature in the m-MTDATA/T2T bilayer can be applied to future excitonic devices[2,42,44].

Figure 5d presents the LCM PL spectra of the m-MTDATA/T2T bilayer at various low temperatures with $\lambda_{ex} = 405$ nm and $P_{in} = 0.1$ μW. The PL peaks at 2.14 eV (=579.4 nm) and 2.90 eV (=427.6 nm) at 290 K correspond to the $XP_{m\text{-}MT/T2T}$ (including $XM_{m\text{-}MT}$) and $XF_{m\text{-}MT}$, respectively. The PL peak positions of $XP_{m\text{-}MT/T2T}$ were blue-shifted from 2.14 eV (290 K) to 2.17 eV (3 K) with decreasing temperature (Fig. 5b) because of the increase in repulsive dipole-dipole interaction between XPs, implying that the delocalized XPs are more aligned at low temperatures with the reduction of fluctuation of dipole moments. The PL peak position (2.90 eV) of $XF_{m\text{-}MT}$ was almost constant over the entire measured temperature range (Fig. 5e), mainly because of the strong binding energy and confinement of the XFs. The LCM PL intensity of $XP_{m\text{-}MT/T2T}$ increased with the temperature increasing from 3 to 80 K and then saturated at above 100 K, whereas that of $XF_{m\text{-}MT}$ remained relatively constant, as shown in Fig. 5f. This behavior of $XP_{m\text{-}MT/T2T}$ correlates with the activation of TADF in D-A m-MTDATA/T2T bilayers[56]. At cryogenic temperatures, nonradiative decay channels are

generally deactivated, suggesting exciton decay in the singlet and triplet states. The transition rate of RISC ($k_{RISC}$) from triplet to singlet states thermally increases; it is expressed as $k_{RISC} = \exp(-\Delta E_{ST}/RT)$, where $\Delta E_{ST}$ is the energy offset between first excited states of singlet and triplet states and $R$ is the universal gas constant[56]. The results presented in Fig. 5f suggest that the $\Delta E_{ST}$ is relatively small at 3.2 meV, causing the easy activation of TADF.

## EL of bilayer (BL) and CDL OLEDs

The potential energy ($U$) of electric dipole moment ($\mathbf{p}$) of exciton species can be modulated via external and local electric fields ($\mathbf{E}_{ext}$ and $\mathbf{E}_{loc}$, respectively) as $U = -\mathbf{p} \cdot (\mathbf{E}_{ext} + \mathbf{E}_{loc})$. This implies that the EL characteristics of OLEDs are modulated by electric fields according to the correlation between the electric fields of the exciton species ($\mathbf{E}_{loc}$) and the external bias ($\mathbf{E}_{ext}$). OLEDs with an m-MTDATA/T2T bilayer as the active layer were fabricated as shown in Fig. 6a. To avoid additional emission, light-emitting materials other than m-MTDATA and T2T were not included in the OLEDs. Figure 6b shows the EL spectra of the m-MTDATA/T2T bilayer OLED with various applied forward biases (1–13 V). The intense EL emissions of $XP_{m\text{-}MT/T2T}$ in the bilayer OLED were observed at 2.13 eV ($V = 9$ V) with the weak EL shoulder peak at 2.93 eV ($V = 9$ V) owing to the $XF_{m\text{-}MT}$. For our bilayer OLEDs, the EL emission of $XP_{m\text{-}MT/T2T}$ was dominant and was considerably enhanced with increasing applied bias ($V \geq 3$ V), suggesting the rapid accumulation of XPs orthogonal to the interface at the HJ with increasing bias. The EL peak of $XF_{m\text{-}MT}$ (at 2.93 eV) was observed at a high applied bias ($V \geq 9$ V) owing to the energy-band offset. The EL peak positions of $XP_{m\text{-}MT/T2T}$ in the bilayer varied distinctively with the applied forward bias compared with those of $XF_{m\text{-}MT}$, as shown in Fig. 6b, c. The blue shift in the EL peak positions with increasing the bias was clearly observed for $XP_{m\text{-}MT/T2T}$ in the bilayer OLED, as shown in Fig. 6c. The direction of the dipole moments of $XP_{m\text{-}MT/T2T}$ was orthogonal to the

interface and opposite to that of the applied electric field by forward bias (Supplementary Fig. 15). With increasing applied forward bias, the concentration of XPs at the interface between D-A (m-MTDATA and T2T) increased, and their repulsive dipole-dipole interaction ($U$) increased. Consequently, the EL peak position was blue-shifted as shown in Fig. 6c. However, the EL peak position of $XF_{m-MT}$ in the same OLED was relatively stable with the applied bias (inset in Fig. 6b) because the dipole moments of $XF_{m-MT}$ were rigidly confined at each m-MTDATA monomer.

OLEDs with m-MTDATA:T2T CDL were fabricated to compare the EL variation owing to XP alignment. m-MTDATA and T2T molecules were simultaneously co-deposited with the same concentration ratio (1:1) in a single layer. The EL spectra of OLED with m-MTDATA:T2T CDL are shown in Supplementary Fig. 16. The EL peak positions (blue markers) of $XP_{m-MT:T2T}$ in the CDL OLED did not change with applied bias, as depicted in the inset of Fig. 6c. Notably, the EL intensities of $XP_{m-MT:T2T}$ for the OLED using the CDL were considerably lower than those using the bilayer, as shown in Fig. 6d. These are attributed to the random (up or down) orientations of the $XP_{m-MT:T2T}$ dipoles along the interfaces between the local m-MTDATA and T2T molecules in the CDL, resulting in an unvarying $U$ with increasing XP density. The LCM PL peak positions of $XP_{m-MT/T2T}$ in the bilayer were blue-shifted, whereas those of $XP_{m-MT:T2T}$ in the CDL remained almost constant (Supplementary Fig. 17) with increasing $P_{in}$. This indirectly supports the distinctive dipole alignment between the bilayer and CDL. Similar behavior of the presence or absence of a blue shift of the EL peak for D-A OLEDs depending on the configuration of molecular orientation was reported by Attar and Monkman[28]. The additional device characteristics such as external quantum efficiency (EQE) and current density-voltage-luminescence (I-V-L) of the bilayer and CDL OLEDs are discussed in the Supplementary Note 16 in the SI. The EQE, current efficiency, and power efficiency of the bilayer and CDL OLEDs are included in Supplementary Figs. 18 and 19.

This study demonstrates the distinctive PL and EL emission characteristics and angular chromism in terms of the dipole directions of XFs and XPs in the m-MTDATA/T2T D-A bilayer and CDL. The ET efficiency between the XP and XF or the XP and TADF dopants depends on the spatial separation of the electron-hole pair for XP and the direction of the dipole moments because of the vector characteristics of the exciton dipole. The Förster resonance energy transfer (FRET) rate is a function of the orientation factor of dipole moments of exciton species, $\kappa^2$ $[\kappa = \boldsymbol{\mu}_D \cdot \boldsymbol{\mu}_A - 3(\boldsymbol{\mu}_D \cdot \mathbf{r}_{DA})(\boldsymbol{\mu}_A \cdot \mathbf{r}_{DA}); \ 0 \leq \kappa^2 \leq 4]$, where $\boldsymbol{\mu}_D$ and $\boldsymbol{\mu}_A$ are the transition dipole moments of the D and A molecules (or two different exciton species such as XF and XP), respectively, and $\mathbf{r}_{DA}$ is the normalized vector connecting centers of $\boldsymbol{\mu}_A$ and $\boldsymbol{\mu}_A$[57]. Therefore, the parallel alignment of the dipole moments of XP, XF, and the dopant molecules can be considered to enhance the quantum efficiency of OLEDs owing to the constructive interference of the electric field of the dipole moments. For example, the efficient ET with the parallel alignment of dipoles between $XP_{m-MT/T2T}$ and $XM_{m-MT}$ (excimers in the m-MTDATA layer) in our m-MTDATA/T2T bilayer enhanced the PL at 575 nm (Fig. 1c and Supplementary Figs. 21 and 22). However, the orthogonality between exciton species can induce a relatively weaker ET based on the factor of $\kappa^2$. The results indicate that both FRET and CT-related Dexter energy transfer effects between parallelly aligned XPs and XMs were involved in the m-MTDATA/T2T bilayer. In contrast, XPs were dominant in the PL spectrum of the m-MTDATA:T2T CDL with a very weak XF PL peak, as shown in Supplementary Fig. 21. The directions of the XF and CT exciton dipole moments in various D-A HSs are also considered for efficient dissociation of the CT excitons caused by the applied backward bias. The directional coupling of the dipole moments of XF and XP are applicable to advanced photonic and optoelectronic devices. Moreover, our results can be applied to excitonic devices using long-lived XP-

polariton formation based on the energy-momentum (E-k) dispersion relations of $XP_{m-MT/T2T}$.

## Discussion

In summary, the characteristics of the PL and EL emissions related to the dipole moments of the intramolecular XF and intermolecular XP (including the excimer) were investigated for the π-conjugated organic D-A m-MTDATA/T2T bilayer and CDL HSs. In the BFP images, the dipole direction of the XPs of the m-MTDATA/T2T bilayer was observed to be out-of-plane and orthogonal to that of the XFs of the m-MTDATA layer. The average lifetime of the XPs corresponding to delayed fluorescence component for m-MTDATA/T2T bilayer was 2.14−4.65 µs (290−3 K), which are considerably longer than that (a few ns) of the XF of m-MTDATA and T2T monomers. In addition, $XP_{m-MT/T2T}$ showed an energy-momentum dispersion relationship, which is clearly distinguishable from that of $XF_{m-MT}$. The angular chromism of the XP of the bilayer was directly observed in the BFP spectra, suggesting delocalization. With increasing excitation power, the PL peak and FWHM of $XP_{m-MT/T2T}$ were blue-shifted and increased, respectively, owing to the enhanced repulsive dipole-dipole alignment of XPs at the interface of the m-MTDATA/T2T bilayer. Intense EL emission was observed at 2.13 eV corresponding to the XPs of the OLEDs using the m-MTDATA/T2T bilayer, and the blue shift of the EL peak positions of XP with increasing applied forward bias originated from the enhanced repulsive dipole alignment of the XPs perpendicular to the interface. These results support the dipole anisotropy of XF and XP in the m-MTDATA/T2T bilayers. Our results and analysis, including the BFP PL spectra, can aid in understanding the dipole-directional coupling of exciton species in organic HSs, which contributes to determining the quantum efficiency of OLEDs and their application in promising photonic devices.

## Methods

### Fabrication of organic layers

m-MTDATA and T2T were used for organic donor (D) and acceptor (A) molecules, respectively. The m-MTDATA powder (purity = 98.0 %) and T2T powder (purity > 98.0%) were purchased from Sigma-Aldrich and TCI, respectively. They were used without further purification. The m-MTDATA and T2T layers, the m-MTDATA/T2T bilayer, and the m-MTDATA:T2T co-deposition layer (CDL) were fabricated using an organic molecular beam deposition (OMBD) system (Daeki Hi-Tech). Further, m-MTDATA and T2T molecules were deposited on to a Si/SiO₂ substrate at a deposition rate of 0.3 nm min⁻¹, and the rate was measured by using a quartz sensor (6 MHz, gold, iTASCO). The deposition was performed in a high-vacuum environment (under $5.0 \times 10^{-6}$ Torr). For co-deposition, the deposition rates were set to be equal, and the molecules were deposited simultaneously.

### Fabrication of organic light-emitting diodes (OLEDs)

Indium tin oxide glass (10 Ohm sq⁻¹) was washed with acetone and isopropyl alcohol and immediately dried using an N₂ gun. Poly(3,4-ethylenedioxythiophene) polystyrene sulfonate (Clevios P VP CH 8000; Heraeus) was used as a hole transport layer. It was spin-coated at 4000 rpm for 60 s, followed by soft baking on a hot plate at 120 °C for 60 s. Subsequently, it was baked in a vacuum oven at 120 °C for 30 min. The m-MTDATA and T2T layers with thicknesses of 20 nm were successively deposited for the bilayer by using an OMBD system at depositions rate of 0.44 and 0.57 nm min⁻¹, respectively. For the CDL OLEDs, the concentration ratio of m-MTDATA and T2T was 1:1. For the cathode, 2 nm of lithium fluoride and 150 nm of aluminum were successively deposited through conventional thermal evaporation in high vacuum environment (under $5.0 \times 10^{-6}$ Torr) at average deposition rates of 0.5 and 0.7 Å s⁻¹, respectively. The fabricated OLED were encapsulated in a glove box with a glass lid to prevent degradation.

## Measurement

Laser confocal microscopy (LCM) photoluminescence (PL) spectra were recorded by using a homemade high-resolution LCM system coupled with a spectrometer (Acton SpectraPro 300i; Princeton Instruments) and a charge-coupled device (CCD; PIXIS 100; Princeton Instruments) at low temperatures (3–290 K) with a closed-loop cryostat (Cryostation; Montana Instruments). A 405-nm diode laser was used for excitation, and a 409-nm long-pass filter was employed to prevent the detector from being exposed to the laser. Time-correlated single-photon counting (TCSPC) systems (Simple-Tau; Becker & Hickl GmbH) were used to record the time-resolved PL (tr-PL) spectra. Further, HPM-100 and PML-16 TCSPC systems were used to determine the single- and 16-channel tr-PL decay curves, respectively. A 375-nm picosecond diode laser (BDL-375-SMN; Becker & Hickl GmbH) was used as the excitation source. For the 16-channel tr-PL measurements, a pulse generator card (DDG-210; Becker & Hickl GmbH), breakout box (BOB-101; Becker & Hickl GmbH), and sync signal generator module (LSG-02; Becker & Hickl GmbH) were used. A 409-nm long-pass filter was used to prevent the detector from being exposed to the laser, and a 514-nm long-pass filter was used in the microsecond-order tr-PL measurements to detect PL from long-lived excitons only. Back focal plane (BFP) PL mapping patterns were obtained using the same LCM system with a customized scanning BFP setup[45]. Ultraviolet photoelectron spectroscopy (UPS) was performed using X-ray photoelectron spectroscopy (XPS) system (Nexsa; Thermo Scientific) at the Korea Institute of Science and Technology. A UV-visible spectroscopy system (Agilent 8453) was used to record the absorption spectra. The electroluminescence (EL) spectra of the OLEDs were determined by using a CCD spectrometer (Andor iDus DV401A-BV) and a source measure unit (Keithley 237).

## Data availability

All data supporting the findings of this study are available from the corresponding author on request.

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

## Acknowledgements

This study was supported by the National Research Foundation (NRF) of Korea funded by the Korean government (No. 2021R1A2C2005885 and No. 2022R1A2C2009412). The work was also supported in part by the NRF under the BK21 FOUR program at Korea University, Initiative for science frontiers on upcoming challenges. Thanks to Prof. D. H. Choi and S. H. Park in Department of Chemistry at Korea University for assisting to measure OLED characteristics.

## Author contributions

J.J. and J.K. conceived the concept and designed the experiments. S.-h.L. synthetically fabricated the D-A HSs and OLEDs and performed low-temperature LCM PL, BFP PL mapping, tr-PL, and EL measurements. T.J.K. and E.L. prepared the tr-PL and low-temperature LCM PL and BFP PL mapping setups. D.K. fabricated D-A HSs. J.J., J.K. and S.-h.L. contributed to data analysis and writing and reviewing of the manuscript. All authors commented on the manuscript.

## Competing interests

The authors declare no competing interests.
