## [Peer Review File · Nature Communications]

Observation of aligned dipoles and angular chromism of exciplexes in organic molecular heterostructuresREVIEWER COMMENTS

Reviewer #1 (Remarks to the Author):

This report examines the dipole orientation of the Frenkel excitons and exciplexes in an organic heterojunction and the effect on emission characteristics of the device. The difference in orientation of the two types of excitons have been mapped using back focal plane photoluminescence. This finding serves as a useful guiding principle in the designing and fabricating devices with enhanced external quantum efficiency, and therefore it is suitable to be published in Nature Communications, provided the following issues are addressed.

1. Please provide device characteristics of OLEDs fabricated and comment on how it correlates with the dipole orientation of excitons.
2. The authors mentioned that both singlet and triplet excitons are involved, so electron transfer between XP and XF cannot be entirely achieved using FRET. Can the author comment on the effect of exciton orientation on the efficiency of Dexter transfer and how it correlates with the emissive characteristics of the devices.
3. Supplementary Fig. 12., the fitting curve doesn't seem to follow the trend of the raw data when the incident power is about $1e-8$ W. Why is that?

Reviewer #2 (Remarks to the Author):

This manuscript reported on the anisotropy emission behavior of exciplexes at the donor /acceptor interfaces such as m-MTDATA and T2T. The interfacial exciplex formation surely provides the transition dipole moment which is perpendicular to the film surface, and the proposed concept and analysis are reasonable. Although such dipole orientation can be anticipated in the interfacial exciplex system, there has been no clear report to prove it. I think the manuscript is worth publishing in NC. I have some requests to improve the description.

- 1) The relationship between the dipole emission scheme and the BFP image is unclear.

Horizontal orientation of dipole----BFP image (intense emission at the center)

Vertical orientation of dipole----BFP image (intense emission at the edges)

It is helpful to provide the theoretical background.

- 2) The orientation parameter (S) of m-MTDATA and T2T films should be measured by using an ellipsometer experimentally and discussing the distribution of dipole orientation.
- 3) The basic transient decay, prompt and delayed components, of the exciplex of m-MTDATA and T2T should be supplied.
- 4) The efficiencies of PLQY, EQE, and outcoupling should be discussed. There is almost no discussion about the efficiencies, providing an unclear overall picture of exciplexes.

Reviewer #3 (Remarks to the Author):

The manuscript „Orientation of aligned dipoles and angular chromism of exciplexes in organic heterostructures” addresses a topic that has become popular due to the appearance of van der Waals heterostructures. In the latter case, interlayer excitons are a distinctly new feature with fascinating properties. However, a simple analogy to the case of charge-transfer excitons – sometimes designated as exciplexes – at organic donor-acceptor interfaces is not that straightforward because of the fundamental differences between structurally well-defined 2D materials, like transition metal dichalcogenides, and disordered molecular semiconductors. And this is one of my main points of criticism: the authors of this manuscript make extensive use of this apparent analogy without considering the different nature of the materials and without any kind of proof that these concepts apply here.

Nevertheless, it is certainly worthwhile studying the different transition dipole orientation of molecular (“Frenkel”) excitons and CT excitons (“exciplexes”) at a donor-acceptor interface. And the finding that they seem to be orthogonal is a priori not surprising. But I have some doubts that the situation depicted in Fig. 3 is really that simple.

First, the two materials (m-MTDATA as donor and T2T as acceptor) show different preferential orientation as evidenced by the opposite sign of their birefringence: m-MTDATA is lying but T2T is standing with respect to the substrate plane.

Second, there is disorder in both the positions and orientations of the molecules so that the interface is certainly not so well-defined but exhibits some roughness, which will lead to a broad distribution of CT exciton orientations and energetics. Thus, I have doubts that concepts of delocalization (p. 11) derived from an apparently parabolic angular dispersion are meaningful. The same problem arises when the apparent blue-shift of the PL spectra are ascribed to “repulsive dipole-dipole interaction” (p. 13). Though it may be tempting to use these analogies to 2D-TMDCs, the validity of these concepts for molecular CT excitons needs to be proven by quantitative calculations, which is not done in this manuscript.

And third, the speculation that CT excitons with longer lifetimes of their excited states (as compared to molecular excitons) are candidates for quantum information processing is quite farfetched. To my understanding, it is not just a long excited-state lifetime that is required, but more importantly a sufficient coherence time.

Now to some technical deficiencies:

- The authors use a 405nm laser for PL excitation in combination with a 409nm longpass filter for detection. However, this filter cuts off a significant fraction of the m-MTDATA emission (Fig. 1). Furthermore, the PL spectrum of this material contains a significant contribution of excimer emission which overlaps with the exciplex emission that is of interest. And even T2T, which has a much larger optical gap, seems to have some PL emission in the range of the exciplex (Fig. S2). These parasitic features are not ideal for a clear assignment of the measured signals.
- The different angular dependence of the exciton and exciplex signals in Fig. 2 are not necessarily an indication of different optical dipole orientation. As explained in Ref. 41, they can both come from in-plane dipoles. As further shown in this Ref., a quantitative analysis of these BFP images is required to come to an unambiguous conclusion.
- It is not true that a CT exciton is more stable than a molecular exciton, as claimed on p. 4. The opposite is the case because the exciton binding energy scales inversely with the distance of the e-h pair.
- The notation $|D^*A\rangle$ etc. does not stand for the wavevector (as claimed on p. 8) but for the wave function of a certain state.
- And what is the reasoning for the parabolic dispersion relation of CT excitons on p. 11?

All in all, the manuscript lacks conciseness and a thorough quantitative analysis to prove the claims.

Response Letter

Reviewer #1's comments

Comments: “This report examines the dipole orientation of the Frenkel excitons and exciplexes in an organic heterojunction and the effect on emission characteristics of the device. The difference in orientation of the two types of excitons have been mapped using back focal plane photoluminescence. This finding serves as a useful guiding principle in the designing and fabricating devices with enhanced external quantum efficiency, and therefore it is suitable to be published in Nature Communications, provided the following issues are addressed.”

Author reply: Thank you for the insightful comments.

1. “Please provide device characteristics of OLEDs fabricated and comment on how it correlates with the dipole orientation of excitons.”

Author reply: Thank you for the helpful comment. To respond to these comments, we measured device characteristics such as current density–voltage–luminance ($I-V-L$), external quantum efficiency (EQE) of OLEDs using BL and CDL of donor m-MTDATA and acceptor T2T. Supplementary Fig. 18a in the revised SI shows $I-V-L$ characteristic curves of the BL (red curves) and CDL (black curves) OLEDs. The BL OLED showed more stable $I-V-L$ properties, whereas the CDL OLED showed decreased luminance and current density with an applied bias over 6.5 V and 9.5 V, respectively. The interlayer junction of the BL and the randomized intermolecular junction of the CDL in the OLEDs induced a difference in device stability, probably because of the relatively homogeneous local electric field near the junction in the BL compared with the inhomogeneous local electric field of the random distribution of the donor m-MTDATA and acceptor T2T molecules in the CDL. The large current of the CDL OLED originates from the easy injection and formation of a hole channel from the ITO to the active layer.

Supplementary Fig. 18. **a** I - V - L curves of BL (red) and CDL (black) OLEDs. **b** EL spectra of BL (red) and CDL (black) OLEDs in a bias of 3.5 V.

Notably, the EL spectrum of the BL OLED corresponding to the XPs was observed at $\lambda_{em} = 585$ nm at a relatively low bias ($=3.5$ V), whereas that of the CDL OLED was very weak, as shown in Supplementary Fig. 18b. This suggests the easy formation of CT excitons with a low bias at the BL interface of the donor m-MTDATA and acceptor T2T.

Supplementary Fig. 19. **a** and **d** EQE, **b** and **e** CE, **c** and **f** PE of BL (red) and CDL (black) OLEDs as functions of bias and current density.

Supplementary Fig. 19 shows the EQE, current efficiency (CE), and power efficiency (PE) as functions of bias and current density for the BL (red) and CDL (black) OLEDs. It is noted that the OLEDs for Supplementary Fig. 19 are new batch devices. The overall efficiency of the BL OLED was higher than that of the CDL OLED. This is because the injected electron-hole recombination rate for radiative decay in the aligned BL OLED is higher than that of the blended D-A molecules in the CDL OLED. The EQE, CE, and PE values of the BL and CDL OLEDs were relatively high in the intermediate range of bias (~ 5 V) and current density (~ 10 mA/cm²). The measured low values of EQE, CE, and PE of the BL and CDL OLEDs can be attributed to low outcoupling by the wide-angle emission of XP, power loss by hole leakage, and/or low charge balancing caused by the lack of functional layers such as the electron injection layer (EIL). Interestingly, a clear correlation between the OLED characteristics and the dipole orientation of excitons/excimeres has been discussed in the manuscript (pages 17–19) in Fig. 6c. The EL peak positions of the BL OLED show a blue shift with increasing applied bias. In contrast, those of the CDL OLED show no shift. This indicates the alignment of XP dipoles with increasing applied bias at the BL of m-MTDATA/T2T.

The photoluminescence quantum yield (PL QY) was measured using an integrating sphere (Newport, 819C-IS-5.3) and a 325-nm-fiber-coupled LED. Supplementary Fig. 20 shows the PL spectra of LED, m-MTDATA/T2T BL (50 nm/50 nm), and m-MTDATA:T2T CDL (1:1, 50 nm). The calculated PL QY values for each sample were 26.4% for XP (with excimers (XM)) for BL and 46.9% for CDL (XP only). These PL QY values for the XPs were similar to those obtained using m-MTDATA and/or T2T [new Supplementary Ref. S6 Applied Physics Letters 80, 2401 (2002); Ref. S7 Advanced Science 6, 1801938 (2019)]. The higher PL QY of the CDL than that of the BL is due to the higher density of CT excitons in the bulky CDL compared to those from the interface of the BL.

Supplementary Fig. 20. PL spectra of 325-nm-fiber-coupled LED (black curves), **a** m-MTDATA/T2T BL (red curve), and **b** m-MTDATA:T2T CDL (red curve).

The above results and discussion of the device characteristics of BL and CDL OLEDs have been included in new section 16 of the revised SI.

2. “The authors mentioned that both singlet and triplet excitons are involved, so electron transfer between XP and XF cannot be entirely achieved using FRET. Can the author comment on the effect of exciton orientation on the efficiency of Dexter transfer and how it correlates with the emissive characteristics of the devices.”

Author reply: Thank you for the helpful comment. As pointed out, electron transfer between XP and XF cannot be entirely achieved using FRET. Dexter energy transfer (DET) induced by charge transfer (CT) can occur during the energy transfer of triplet states. In response to the reviewer’s comment, we have included the following discussion in the revised manuscript and SI (section 17. Comparison of LCM PL spectra...): “As shown in Supplementary Fig. 21a and b, PL peak of XPs in m-MTDATA/T2T bilayer was considerably enhanced, decreasing that of XM, while PL peak of XF was weakly changed. As shown in Supplementary Fig. 3a, there was almost no characteristic absorption peak in the range of 450–550 nm after the hybridization of m-MTDATA and T2T. These results indicate that both FRET and CT-related DET effects between parallelly aligned XPs and XMs are involved in the MTDATA/T2T BL. In contrast, XPs were dominant in the PL spectrum of the m-MTDATA:T2T CDL with a very weak XF PL peak, as shown in new Supplementary Fig. 21c. This suggests that the DET effect was relatively dominant for the D-A CDL because of the much larger overlap of wave functions from the random and

opposite dipole moments of XPs from the blending donor and acceptor molecules. In terms of OLED performance, the EL peaks of XP for the BL OLED were rapidly enhanced and blue-shifted with increasing applied bias. In contrast, those of the CDL OLED were very weak and not shifted, as shown in Fig. 6 and Supplementary Fig. 16. This originates from the different DET efficiencies and configurations of the dipole orientations of XPs, XMs, and XPs in our organic m-MTDATA and T2T BL and CDL systems.”

Supplementary Fig. 21 LCM PL spectra of m-MTDATA layer and m-MTDATA/T2T BL at **a** 3 K and **b** 290 K. **c.** LCM PL spectrum of m-MTDATA:T2T CDL at 3 K.

3. “Supplementary Fig. 12., the fitting curve doesn’t seem to follow the trend of the raw data when the incident power is about 1e-8 W. Why is that?”

Author reply: Thank you for the comment. We examined the PL spectra obtained at various excitation powers. We found that, as shown in Fig. R1a, b, and c, the signal-to-noise ratios of PL spectra of m-MTDATA/T2T BL with a $P_{in} = 5.0 \times 10^{-9}$ W, 7.0×10^{-9} W, and 9.0×10^{-9} W, respectively, were quite low compared to those obtained with higher powers, as shown in Fig. R1d ($P_{in} = 1.0 \times 10^{-6}$ W) as an example, questioning the reliability of our previous assignment of the peak position of XP from the PL spectra obtained with an excitation power less than $P_{in} = 1.0 \times 10^{-8}$ W. We note that the power range that we used expands more than three orders of magnitude; therefore, we ruled out the PL peak positions below $P_{in} = 1.0 \times 10^{-8}$ W for the fitting, as shown in new Supplementary Fig. 17.

Fig. R1. LCM PL spectra and fitting curves of m-MTDATA/T2T bilayer with **a** $P_{in} = 5.0$ nW, **b** 7.0 nW, **c** 9.0 nW, and **d** $P_{in} = 1.0$ μ W.

New Supplementary Fig. 17. Incident power dependence of PL peak positions of XPs from m-MTDATA/T2T BL (black) and m-MTDATA:T2T CDL (red) at 3 K. The dotted lines are eye-guided lines.

Reviewer #2's comments

Comments: “This manuscript reported on the anisotropy emission behavior of exciplexes at the donor /acceptor interfaces such as m-MTDATA and T2T. The interfacial exciplex formation surely provides the transition dipole moment which is perpendicular to the film surface, and the proposed concept and analysis are reasonable. Although such dipole orientation can be anticipated in the interfacial exciplex system, there has been no clear report to prove it. I think the manuscript is worth publishing in NC. I have some requests to improve the description.”

Author reply: Thank you for the positive comments.

1. “The relationship between the dipole emission scheme and the BFP image is unclear. Horizontal orientation of dipole----BFP image (intense emission at the center)
Vertical orientation of dipole----BFP image (intense emission at the edges)
It is helpful to provide the theoretical background.”

Author reply: Thank you for the helpful comment. As the reviewer suggested, the interpretation of BFP images to identify the exciton dipole orientations has been newly added on Page 9 as the following: “In BFP experiments, the more intense emission is expected at the center of BFP than toward the edge of BFP owing to low angle emission from the horizontal (in-plane) orientation of dipoles for XFs. With a similar analogy, a relatively more intense emission can be detected toward the edge of the BFP than at the center of the BFP owing to the high-angle emission from the vertical (out-of-plane) orientation of dipoles of XPs.”

2. “The orientation parameter (S) of m-MTDATA and T2T films should be measured by using an ellipsometer experimentally and discussing the distribution of dipole orientation.”

Author reply: Thank you for your insightful comments. As the reviewer suggests, the ordinary (n_o) and extraordinary (n_e) refractive indices of m-MTDATA, T2T molecules, and their BL as a function of wavelength were calculated using DeltaPsi2 software from the measured data of variable-angle spectroscopic ellipsometry (VASE) using an ellipsometer (UVISEL Plus, HORIBA), as shown in new Supplementary Fig. 9. The n_o and n_e values of the m-MTDATA layer were calculated to be 1.72 and 1.70 (negative

birefringence) at $\lambda = 550$ nm, which agrees with previous results [new Ref. 49. Salehi, A., et al. Recent Advances in OLED Optical Design. *Adv. Func. Mater.* **29**, 1808803 (2019)]. These results indicated that the m-MTDATA molecules had a stacked in-plane columnar structure, agreeing with the results in Ref. 39. The calculation of n_o and n_e values of the T2T layer and m-MTDATA/T2T BL using OMBD should be considered more complicated because of the nanorod structure of the T2T molecules and their random deposition, as shown in Fig. 3a. Because the deposited configuration of the T2T molecules is non-uniform, the model of the exponential gradient layer with anisotropy and vacuum defects (voids) was used, which comprises the bottom and top layers with ordinary and extraordinary terms, respectively. The calculation result of the T2T bottom layer of the m-MTDATA/T2T BL gave the n_o and n_e values as 1.59 and 1.0 (negative birefringence) at $\lambda = 550$ nm, respectively. The calculation result of the T2T top layer of the BL gave the n_o and n_e values as 1.04 and 1.23, respectively, (positive birefringence). The relatively lower values of n_o and n_e for the BL compared to a single layer of previous results [new Ref. 49. Salehi, A., et al. Recent Advances in OLED Optical Design. *Adv. Func. Mater.* **29**, 1808803 (2019)] might be due to the random orientation of the T2T nanorods and the large air space between them.

Supplementary Fig. 9. Calculated refractive indices of **a** m-MTDATA layer and **b** T2T on m-MTDATA layer as functions of wavelength obtained from spectroscopic ellipsometry measurements.

The cross-sectional SEM and TEM images can support the analysis of ellipsometer calculations. The cross-sectional SEM images of the m-MTDATA, T2T, and m-MTDATA/T2T BL are shown in Supplementary Fig. 10. The m-MTDATA/T2T BLs were different batch samples with greater thicknesses. The cross-sectional SEM image

confirmed the in-plane columnar stacking for the m-MTDATA layer (Supplementary Fig. 10a), whereas a random distribution of T2T molecules was observed (Supplementary Fig. 10b). Notably, the T2T molecules were stacked in-plane on the surface of m-MTDATA (i.e., at the interface) and then stacked with randomly standing nanorods on the outer surface, as shown in Supplementary Fig. 10c. The cross-sectional SEM and TEM images clearly demonstrate distinctive molecular stacking in the m-MTDATA/T2T BL. Interestingly, the in-plane interface between the m-MTDATA and T2T layers was clearly observed (Supplementary Fig. 11a and b), whereas there was not clear interface for the CDL as shown in Supplementary Fig. 11c and d.

Supplementary Fig. 10 Cross-sectional SEM images of **a** m-MTDATA, **b** T2T, and **c** m-MTDATA/T2T BL.

Supplementary Fig. 11 Magnification of cross-sectional SEM and TEM images of (a and b) BL and (c and d) CDL of m-MTDATA and T2T. Yellow curves represent intensity profiles of corresponding TEM images.

In sum, for the m-MTDATA/T2T BL, the results of ellipsometry experiments combined with the cross-sectional SEM and TEM images suggest that m-MTDATA and T2T molecules were in-plane stacking at the interface, above which the T2T molecules were randomly stacked out-of-plane at the top as the form of nanorods, as shown in Fig. 3a and Supplementary Fig. 10 and 11. The results of the ellipsometer experiments and cross-sectional SEM and TEM images are included in new section 9 of the revised SI.

3. “The basic transient decay, prompt and delayed components, of the exciplex of m-MTDATA and T2T should be supplied.”

Author reply: Thank you for the helpful comment. We have included the following discussion in the section 12 of the revised SI.

Time-resolved PL (tr-PL) decay curves were analyzed by fitting them to the bi-exponential decay function $y(t) = a_1 \exp\left[-\frac{t}{\tau_1}\right] + a_2 \exp\left[-\frac{t}{\tau_2}\right]$,

The averaged lifetimes were calculated using the intensity-weighted average lifetime, $\tau_{\text{avg}} = \Sigma(a_i \tau_i^2) / \Sigma(a_i \tau_i)$ ($i = 1, 2$). The fitting results are presented in Supplementary Table 2.

Supplementary Table 2. Fitting results of tr-PL decay curves for the prompt and delayed components of XP at various temperatures

Prompt component of XP							
temperature (K)	3	50	100	150	200	250	290
a_1	0.59	0.50	0.54	0.52	0.47	0.47	0.34
τ_1 (ns)	1.49	1.55	1.26	1.31	1.19	1.06	0.93
a_2	0.41	0.50	0.46	0.48	0.53	0.53	0.66
τ_2 (ns)	7.54	8.41	6.36	5.95	4.85	4.23	3.74
τ_{avg} (ns)	6.20	7.33	5.39	5.06	4.19	3.66	3.43
Delayed component of XP							
temperature (K)	3	50	100	150	200	250	290
a_1	0.37	0.55	0.57	0.55	0.58	0.63	0.66

τ_1 (μs)	1.34	1.56	1.21	1.12	1.15	1.05	0.82
a_2	0.63	0.45	0.43	0.45	0.42	0.37	0.34
τ_2 (ns)	5.15	5.40	4.81	4.49	4.54	3.94	2.87
τ_{avg} (μs)	4.65	4.41	3.90	3.69	3.67	3.04	2.14

The following methods were used to measure the prompt and delayed components of the exciplex in the tr-PL experiment.

Fig. R2. Description of tr-PL measurement sequence.

In our tr-PL measurement system, prompt and delayed fluorescence are measured in two consecutive steps, as the process are described in Fig. R2. In the first step (Step 1), the excitation laser pulses at 25 MHz are illuminated on the sample and synchronized electric signal pulse counts the PL photons with their arrival time recorded up within the 40 ns period. With this process repeated for the given integration time (usually 10s of seconds) the complete tr-PL prompt decay curve is obtained. Then, the second step (Step 2) starts with the excitation laser turned off and the PL intensity is detected in 40 ns increment pulse to record the delayed PL decay response up to a few 10s of μs .

During our revision, we found an error in Fig. 5b and corrected one data point of τ_{avg} at 3 K Fig. 5b as the following figure.

New Fig. 5b. Original (left) and corrected figures (right).

4. “The efficiencies of PLQY, EQE, and outcoupling should be discussed. There is almost no discussion about the efficiencies, providing an unclear overall picture of exciplexes.”

Author reply: Thank you for the helpful comment. To respond to these comments, we measured device characteristics such as current density–voltage–luminance (I – V – L), external quantum efficiency (EQE) of OLEDs using BL and CDL of donor m-MTDATA and acceptor T2T. Supplementary Fig. 18a in the revised SI shows I – V – L characteristic curves of the BL (red curves) and CDL (black curves) OLEDs. The BL OLED showed more stable I – V – L properties, whereas the CDL OLED showed decreased luminance and current density with an applied bias over 6.5 V and 9.5 V, respectively. The interlayer junction of the BL and the randomized intermolecular junction of the CDL in the OLEDs induced a difference in device stability, probably because of the relatively homogeneous local electric field near the junction in the BL compared with the inhomogeneous local electric field of the random distribution of the donor m-MTDATA and acceptor T2T molecules in the CDL. The large current of the CDL OLED originates from the easy injection and formation of a hole channel from the ITO to the active layer.

Supplementary Fig. 18. **a** I – V – L curves of BL (red) and CDL (black) OLEDs. **b** EL

spectra of BL (red) and CDL (black) OLEDs in a bias of 3.5 V.

Notably, the EL spectrum of the BL OLED corresponding to the XPs was observed at $\lambda_{em} = 585$ nm at a relatively low bias ($=3.5$ V), whereas that of the CDL OLED was very weak, as shown in Supplementary Fig. 18b. This suggests the easy formation of CT excitons with a low bias at the BL interface of the donor m-MTDATA and acceptor T2T.

Supplementary Fig. 19. a and d EQE, b and e CE, c and f PE of BL (red) and CDL (black) OLEDs as functions of bias and current density.

Supplementary Fig. 19 shows the EQE, current efficiency (CE), and power efficiency (PE) as functions of bias and current density for the BL (red) and CDL (black) OLEDs. It is noted that the OLEDs for Supplementary Fig. 19 are new batch devices. The overall efficiency of the BL OLED was higher than that of the CDL OLED. This is because the injected electron-hole recombination rate for radiative decay in the aligned BL OLED is higher than that of the blended D-A molecules in the CDL OLED. The EQE, CE, and PE values of the BL and CDL OLEDs were relatively high in the intermediate range of bias (~ 5 V) and current density (~ 10 mA/cm²). The measured low values of EQE, CE, and PE of the BL and CDL OLEDs can be attributed to low outcoupling by the wide-angle emission of XP, power loss by hole leakage, and/or low charge balancing caused by the

lack of functional layers such as the electron injection layer (EIL). Interestingly, a clear correlation between the OLED characteristics and the dipole orientation of excitons/exciplexes has been discussed in the manuscript (pages 17–19) in Fig. 6c. The EL peak positions of the BL OLED show a blue shift with increasing applied bias. In contrast, those of the CDL OLED show no shift. This indicates the dipole alignment of the XPs with increasing applied bias at the BL of m-MTDATA and T2T.

The photoluminescence quantum yield (PL QY) was measured using an integrating sphere (Newport, 819C-IS-5.3) and a 325-nm-fiber-coupled LED. Supplementary Fig. 20 shows the PL spectra of LED, m-MTDATA/T2T BL (50 nm/50 nm), and m-MTDATA:T2T CDL (1:1, 50 nm). The calculated PL QE values for each sample were 26.4% for XP (with excimers (XM)) for BL and 46.9% for CDL (XP only). These PL QY values for the XPs were similar to those obtained using m-MTDATA and/or T2T [new Supplementary Ref. S6. Applied Physics Letters 80, 2401 (2002); Ref. S7. Advanced Science 6, 1801938 (2019)]. The higher PL QY of the CDL than that of the BL is due to the higher density of CT excitons in the bulky CDL compared to those from the interface of the BL.

Supplementary Fig. 20. PL spectra of 325-nm-fiber-coupled LED (black curves), **a** m-MTDATA/T2T BL (red curve), and **b** m-MTDATA:T2T CDL (red curve).

The above results and discussion of the device characteristics of BL and CDL OLEDs have been included in new section 16 of the revised SI.

Reviewer #3's comments

Comments: “The manuscript “Orientation of aligned dipoles and angular chromism of exciplexes in organic heterostructures” addresses a topic that has become popular due to

the appearance of van der Waals heterostructures. In the latter case, interlayer excitons are a distinctly new feature with fascinating properties. However, a simple analogy to the case of charge-transfer excitons – sometimes designated as exciplexes – at organic donor-acceptor interfaces is not that straightforward because of the fundamental differences between structurally well-defined 2D materials, like transition metal dichalcogenides, and disordered molecular semiconductors. And this is one of my main points of criticism: the authors of this manuscript make extensive use of this apparent analogy without considering the different nature of the materials and without any kind of proof that these concepts apply here.”

Author reply: Thank you for the comment. As pointed out by the reviewer, there are fundamental differences in the structural and interfacial properties of TMDC-based heterostructures (HSs) and organic D-A molecular HSs. We agree with the rough interface of organic D-A molecular HSs. However, the results of 2D GIWAXS of the physically deposited m-MTDATA in Ref. 39, “the highly organized liquid-phase crystalline stacking of m-MTDTA” were reported, indicating not heavily disordered systems. To investigate the layer structural and interfacial characteristics, we measured the cross-sectional SEM and TEM images of the m-MTDATA, T2T, and m-MTDATA/T2T BL and CDL layers (Supplementary Fig. 10 and 11 in the revised SI). The m-MTDATA/T2T BL and CDL were different batch samples with greater thicknesses. The in-plane columnar stacking was confirmed for the m-MTDATA layer from the cross-sectional SEM images, which was supported by the negative birefringence. Notably, the T2T molecules were mainly stacked in-plane on the surface of the m-MTDATA and then stacked with randomly standing nanorods on the outer surface, as shown in Supplementary Fig. 10c. The cross-sectional SEM and TEM images of the m-MTDATA/T2T BL clearly demonstrate a layered stacking to generate XPs. Interestingly, the interface (i.e., boundary) between the m-MTDATA and T2T D-A layers was clearly observed, whereas there was not clear interface for the CDL, as shown in Supplementary Fig. 11.

Supplementary Fig. 10. Cross-sectional SEM images of **a** m-MTDATA, **b** T2T and **c** m-MTDATA/T2T BL.

Supplementary Fig. 11 Magnification of cross-sectional SEM and TEM images of **(a** and **b)** BL and **(c** and **d)** CDL of m-MTDATA and T2T, respectively. Yellow curves of corresponding TEM images represent the electron scattering intensity profiles.

CT excitons in HSs can generally be formed in type-II band structures, which have been observed not only in TMDC-based HSs (e.g., MoS₂/WSe₂, MoSe₂/WSe₂, etc.) and WS₂/PbI₂ (Ref. 10) but also in other type-II band structures constituting various semiconducting quantum materials, such as TMDC/perovskite (WSe₂/(iso-BA)₂PbI₄) HS (new Ref. 11), perovskite/QD (MAPbI₃/CdS-ZnSe-QD) HS (Ref. 12), and PbS-CdS-QD/MAPbI_{3-x}Cl_x HS (new Ref. 13). Notably, the HSs of the perovskite/QDs also exhibited rough interfaces and the formation of CT excitons. For organic D-A systems, CT excitons such as hetero-intermolecular excitons; exciplexes (XPs) with type-II HOMO-LUMO energy levels have already been studied for application in OLEDs (Refs. 19, 26–29). In our study, we estimated the HOMO and LUMO energy levels of the m-MTDATA and T2T molecules using UPS and optical absorption spectra and confirmed the type-II band structure, as shown in Fig. 1d and Supplementary Fig. 3. Photo-induced CT effects were observed in the PL spectra of XFs, XMs, and XPs, as shown in Fig. 1c.

Therefore, even with the organic molecular stackings and rough interfaces of organic m-MTDATA/T2T systems, the formation of CT excitons has been experimentally confirmed. This is similar to the case of perovskites/QDs also with rough interfaces. For perovskites/QDs (diameter of QDs = 2-10 nm), the elemental mapping of EDX showed rough interfaces and an inhomogeneous distribution of QDs (diameter of QDs = 2–10 nm) on the surfaces of the perovskites (Fig. R3, Ref. 12 and new Ref. 13). Despite rough interfaces and some possible disordering, robust existence of CT excitons (IX and XPs) has been observed in various inorganic and organic semiconductor-based HSs with type-II energy band structures.

New Ref. 11 Y. Chen et al., Robust Interlayer Coupling Two-Dimensional Perovskite/Monolayer Transition Metal Dichalcogenide Heterostructures, *ACS Nano* **14**, 10258 (2020).

New Ref. 13, Sanchez et al., Tunable light emission by exciplex state formation between hybrid halide perovskite and core/shell quantum dots: Implications for advanced LEDs and photovoltaics, *Sci. Adv.* **2**, e15011104 (2016).

Fig. R3. **a** TEM and **b** EDX of MAPbI₃/CdSe-ZnS-QDs HS [12, 13].

For comparison, we obtained cross-sectional TEM and SEM images of the BL and CDL of m-MTDATA and T2T, as shown in Supplementary Fig. 10 and 11. The average sizes of m-MTDATA and T2T molecules were 1–2 nm in the planar direction. From the cross-sectional TEM and SEM images, we can clearly observe the interface of the m-MTDATA donor and T2T acceptor layers, as shown in Supplementary Fig. 11. SEM and TEM images and the interface conditions of the BL and CDL are included in the revised SI.

In the Introduction (Page 3) of the revised manuscript, we have included the features of CT excitons in various inorganic and organic semiconductor-based HSs with

new references as the following: “The CT excitons in heterostructures (HSs) can be generally formed in type-II band structure, which have been observed in not only TMDC-based HSs (e.g., MoS₂/WSe₂, MoSe₂/WSe₂, etc.) and WS₂/PbI₂ (Ref. 10) but also other various type-II band hetero-structures such as TMDC/perovskite (WSe₂/(iso-BA)₂PbI₄) HS (new Ref.11), perovskite/quantum dot (MAPbI₃/CdS-ZnSe-QD) HS (Ref. 12), and PbS-CdS-QD/MAPbI_{3-x}Cl_x HS (new Ref. 13). Notably, the formation of CT excitons has been observed for the HSs of perovskites/QDs with rough interfaces (Ref. 12 and new Ref. 13). The long lifetime and directional characteristics of electric dipole moments of the CT excitons were reported in the perovskite/QDs HSs (Ref. 12 and new Ref. 13).”

“Nevertheless, it is certainly worthwhile studying the different transition dipole orientation of molecular (“Frenkel”) excitons and CT excitons (“exciplexes”) at a donor-acceptor interface. And the finding that they seem to be orthogonal is a priori not surprising. But I have some doubts that the situation depicted in Fig. 3 is really that simple.”

Author reply: Thank you for the comment. To respond to the comment that “the finding that they seem to be orthogonal is a priori not surprising,” the importance of our study was captured by Reviewer 2 as “Although such dipole orientation can be anticipated in the interfacial exciplex system, there has been no clear report to prove it. I think the manuscript is worth publishing in NC.”

As the Reviewer pointed out, Fig. 3 in the original submission was too simple. Based on the results of the ellipsometer experiments and the cross-sectional SEM and TEM images, Fig. 3 has been revised as follows:

- **New Fig. 3.** **a** Cross-sectional SEM image of m-MTDATA/T2T BL. **b** Schematic of stacking of m-MTDATA and T2T molecules for bilayer deposition. Schematic of

dipole formation and distinctive emission directions of $\text{XF}_{\text{m-MT}}$ and $\text{XP}_{\text{m-MT/T2T}}$ in m-MTDATA/T2T bilayer.

“First, the two materials (m-MTDATA as donor and T2T as acceptor) show different preferential orientation as evidenced by the opposite sign of their birefringence: m-MTDATA is lying but T2T is standing with respect to the substrate plane.”

Author reply: Thank you for the helpful comment. As the reviewer suggests, the ordinary (n_o) and extraordinary (n_e) refractive indices of m-MTDATA, T2T molecules, and their BL as a function of wavelength were calculated using DeltaPsi2 software from the measured data of variable-angle spectroscopic ellipsometry (VASE) using an ellipsometer (UVISEL Plus, HORIBA), as shown in new Supplementary Fig. 9. The n_o and n_e values of the m-MTDATA layer were calculated to be 1.72 and 1.70 (negative birefringence) at $\lambda = 550$ nm, which agrees with previous results [new Ref. 49. Salehi, A., et al. Recent Advances in OLED Optical Design. *Adv. Func. Mater.* **29**, 1808803 (2019)]. These results indicated that the m-MTDATA molecules had a stacked in-plane columnar structure, agreeing with the results in Ref. 39. The calculation of n_o and n_e values of the T2T layer and m-MTDATA/T2T BL using OMBD should be considered more complicated because of the nanorod structure of the T2T molecules and their random deposition, as shown in Fig. 3a. Because the deposited configuration of the T2T molecules is non-uniform, the model of the exponential gradient layer with anisotropy and vacuum defects (voids) was used, which comprises the bottom and top layers with ordinary and extraordinary terms, respectively. The calculation result of the T2T bottom layer of the m-MTDATA/T2T BL gave the n_o and n_e values as 1.59 and 1.0 (negative birefringence) at $\lambda = 550$ nm, respectively. The calculation result of the T2T top layer of the BL gave the n_o and n_e values as 1.04 and 1.23, respectively, (positive birefringence). The relatively lower values of n_o and n_e for the BL compared to a single layer of previous results [new Ref. 49. Salehi, A., et al. Recent Advances in OLED Optical Design. *Adv. Func. Mater.* **29**, 1808803 (2019)] might be due to the random orientation of the T2T nanorods and the large air space between them. For m-MTDATA/T2T BL, the results of ellipsometry experiments combined with the cross-sectional SEM and TEM images suggest that m-MTDATA and T2T molecules were in-plane stacking at the interface, above which the T2T molecules were randomly stacked out-of-plane at the top as the form of nanorods, as shown in Fig. 3 and Supplementary Fig. 10 and 11.

Supplementary Fig. 9. Calculated refractive indices of **a** m-MTDATA layer and **b** T2T layer on m-MTDATA as functions of wavelength obtained by spectroscopic ellipsometry measurements.

“Second, there is disorder in both the positions and orientations of the molecules so that the interface is certainly not so well-defined but exhibits some roughness, which will lead to a broad distribution of CT exciton orientations and energetics. Thus, I have doubts that concepts of delocalization (p. 11) derived from an apparently parabolic angular dispersion are meaningful. The same problem arises when the apparent blue-shift of the PL spectra are ascribed to “repulsive dipole-dipole interaction” (p. 13). Though it may be tempting to use these analogies to 2D-TMDCs, the validity of these concepts for molecular CT excitons needs to be proven by quantitative calculations, which is not done in this manuscript.”

Author reply: Thank you for the comment. We measured the cross-sectional SEM and TEM images to respond to the disorder in both the positions and orientation and the not-so-well-defined interface. The interface between the donor m-MTDATA and acceptor T2T layers was observable, and the D-A molecules were stacked quietly at the interface, as shown in Supplementary Fig. 10 and 11. This information has been included in new section 9 in the revised SI. However, we emphasize that the observed parabolic angular dispersion is the result of delocalization specific only to XP formed in BL or CDL, which are further proven with our additional experiments and analysis as discussed in detail as following.

Regarding charge localization and delocalization in π -conjugated conducting polymers and organic small-molecule films, charges can be localized with increasing lattice disorder and structural imperfections. Based on the BFP results, we now discuss

the momentum (k)-dependent PL spectra of m-MTDATA, T2T BL, and CDL. The bilayer (BL) has a more extended interface between the donor and acceptor layers, than those in the CDL that are spatially randomly distributed. As we described on Page 11, “Moreover, the emission energy of XP_{m-MT/T2T} exhibited an obvious angular dispersion, where the emission peak position of XP_{m-MT/T2T} was blue-shifted as $|k_x/k_0|$ increased, implying the energy-momentum dispersion characteristics.” We analyzed the PL energy dispersion with measured momentum as the “delocalization nature (on Page 11),” because for quantum particles, the degree of wavefunction delocalization or spatial extension appears as such energy-momentum dispersion. Exciton–polariton hybridization between excitons and infinitely delocalized photons is one example [Ref. R1,R2], and the energy-momentum dispersion of the exciton–polariton is a direct indication of successful hybridization between excitons and photons. To avoid the misleading of “delocalization,” we have included an additional explanation (on page 11) as “The delocalization here represents relatively broad overlap of wavefunctions between aligned dipoles of XPs in the bilayer due to the large spatial extension of XP wavefunction.” In addition, as shown in Fig. 2g, less dispersive momentum (k)-dependent PL spectra were observed for the XPs of CDL, which were more localized than the BL XPs because of CDL's randomly distributed molecular interfaces. As shown in Supplementary Fig. 7 and 8 in the revised SI, new samples were prepared to confirm our results.

As shown in Fig. 2f and 2g, momentum dispersion of XP was observed specifically in the BL and CDL samples. To investigate the relationship between the XP energy-momentum dispersion and the degree of dipole alignment of the XPs, we prepared a more disordered donor and acceptor blended film and performed the BFP measurement. Supplementary Fig. 7c shows E vs. k of the drop-cast film of the blended m-MTDATA-T2T molecules. Supplementary Fig. 7d shows E vs. k of the reprecipitated film of the blended m-MTDATA-T2T molecules. The drop-cast and the reprecipitated films could be treated as heavily disordered molecular systems, which were made by drop-casting on a hotplate at 120 °C and injecting in deionized water (with vigorous stirring) of the tetrahydrofuran (THF) solution with m-MTDATA and T2T mixture (weight ratio of m-MTDATA and T2T is 1:1), respectively, on Si/SiO₂ substrates. Although the photon energy of XP is identical to that of our BL, the E vs. k relations of the XP peak from the BFP PL spectra of the drop-cast and reprecipitated films show nondispersive flat characteristics because of the severely random distribution of XP dipole orientations. This control experiment results confirm that the observed E - k dispersion characteristics of the BL originate from the broad overlap of the wave functions (delocalization) of the XPs in the BL. Their dispersive characters are quantitatively expressed using the estimated

curvature $\kappa_{\text{center}} \equiv d^2E/dk^2$ in the Supplementary Table 1. The values of κ_{center} of disordered m-MTDATA:T2T blended films of drop-cast or reprecipitated films are negligible compared to κ_{center} of BL and CDL, showing no dispersive characteristics of E vs. k relation. These have been included in new section 7 in the revised SI.

Supplementary Fig. 7. Energy–momentum (E – k) dispersion of **a** BL, **b** CDL, **c** drop-cast film of m-MTDATA-T2T blended molecules, **d** reprecipitated film of m-MTDATA-T2T blended molecules. White dashed lines are parabolic fitting curves.

Supplementary Table 1. κ_{center} values of XP in various HS samples

Sample	BL	CDL	Drop-cast film	Reprecipitated film
κ_{center}	0.499	0.259	0.023	0.055

We note that blue shift of the PL for IXs similar to observed blue shift of XP peak energy in our sample was also reported not only in TMDC bilayers but also in other kinds of HS, such as perovskite/TMDC ($\text{WSe}_2/(\text{iso-BA})_2\text{PbI}_4$) in a new Ref. 11 and $\text{MAPbI}_3/\text{CdS-ZnSe-QD}$ HS (Ref. 12) that supposedly have imperfect (or rough) interfaces. The relationship between the PL blue shift and repulsive dipole interaction has

been theoretically and experimentally investigated in the indirect excitons of coupled quantum wells (new Refs. 52 and 53).

New Ref. 11, Chen, Y. et al. Robust interlayer coupling in two-dimensional perovskite/monolayer transition metal dichalcogenide heterostructures. *ACS Nano* **14**, 10258–10264 (2020).

New Ref. 52, Liu, C. S., et al. Theory of indirect exciton photoluminescence in elevated quantum trap. *Physica E: Low-Dimen. Syst. Nanostructures* **63**, 193–198 (2014).

New Ref. 53, Choksy, D. J, et al., Attractive and repulsive dipolar interaction in bilayers of indirect excitons. *Phys. Rev. B* **103**, 045126 (2021).

Ref. R1, Michalsky, T., et al. Spatiotemporal Evolution of Coherent Polariton Modes in ZnO Microwire Cavities at Room Temperature. *Nano Letters*, **18**, 6820–6825 (2018).

Ref. R2, Randy P. S., et al. Organic polariton lasing with molecularly isolated perylene diimides. *Appl. Phys. Lett.* **117**, 041103 (2020).

Our analysis for the PL blue-shift due to the enhanced repulsive dipole-dipole interaction in the D-A molecular HSs is highly consistent with previous reports of IXs in various HSs discussed above. With increasing power of excitation laser, the concentration of accumulated charges at the heterojunction of D-A interface increased, resulting in the increase of dipoles corresponding to XPs. This induced the enhanced repulsive dipole-dipole interaction. Even though the roughness of interface in organic D-A molecular systems, it was observed that the ensemble (i.e., net) dipole moments of XPs was out-of-plane from the BFP imaging. These have been included on page 15 with new Refs. 11, 52, and 53 in the revised manuscript as the following: “With increasing power of excitation laser, the concentration of accumulated charges at the heterojunction of D-A interface increased, resulting in the increase of density of dipoles corresponding to XPs and enhancing the repulsive dipole-dipole interaction. The similar results have been observed in other perovskite/TMDC ($\text{WSe}_2/(\text{iso-BA})_2\text{PbI}_4$) (new Ref.11) and $\text{MAPbI}_3/\text{CdS-ZnSe-QD}$) HS (Ref. 12) with their rough interfaces. The relationship between PL blue-shift and repulsive dipole interaction had been theoretically and experimentally studied in indirect excitons of coupled quantum wells (new Refs. 52 and 53)”.

Quantitative analysis of blue-shift of XPs or IXs with increasing density of exciton dipoles requires the quantum mechanical calculation of Coulomb interactions between aligned dipoles considering the proper wavefunction parameters of XPs, distribution of XP orientations, dielectric environment etc. Inspired by the reviewer’s suggestion, we believe that such theoretical studies are certainly an interesting subject for future study

but are beyond the scope of our current study that focused on the solid experimental demonstration of unique XP features of all-organic HSs.

“And third, the speculation that CT excitons with longer lifetimes of their excited states (as compared to molecular excitons) are candidates for quantum information processing is quite farfetched. To my understanding, it is not just a long excited-state lifetime that is required, but more importantly a sufficient coherence time.”

Author reply: Thank you for the comment. As the reviewer suggested, we have deleted the words of “quantum information processing” in this revised manuscript.

Now to some technical deficiencies:

1. “The authors use a 405nm laser for PL excitation in combination with a 409nm long pass filter for detection. However, this filter cuts off a significant fraction of the m-MTDATA emission (Fig. 1). Furthermore, the PL spectrum of this material contains a significant contribution of excimer emission which overlaps with the exciplex emission that is of interest. And even T2T, which has a much larger optical gap, seems to have some PL emission in the range of the exciplex (Fig. S2). These parasitic features are not ideal for a clear assignment of the measured signals.”

Author reply: Thank you for your careful review. We show in Fig. R3 the PL spectrum of the m-MTDATA/T2T BL without the 409-nm long-pass (LP) filter cut-off (black curve) obtained with much reduced intensity of excitation laser of 405 wavelength to avoid the damage of our CCD detector. The comparison to the PL with the LP cut-off showed that the loss by the LP filter in the m-MTDATA spectrum is negligible.

Fig. R3. LCM PL spectra of m-MTDATA/T2T BL with (red) and without (black) 409-

nm long-pass filter. Note that the laser excitation power had to be substantially reduced because of the absence of the long-pass filter and strong laser line detection. Therefore, the apparent S/N of PL spectrum for the no filter case is substantially lower than that of LP=409 nm case.

As the reviewer has pointed out, we agree that the PL spectra of m-MTDATA in the BL include the emission by XM. However, we show below that our key interpretation of XP in terms of angular dispersion and aligned dipoles shown in BFP imaging are not affected by the partial inclusion of XM in PL spectra of m-MTDATA/T2T BL or CDL. In Fig. R4 (left), we show the PL spectrum of BL subtracted by m-MTDATA spectrum with the normalizing adjustment of PL peak height. Adjustment was performed by matching the high-energy edge curve of the XM signal in the m-MTDATA spectrum with that of the XM-including XP signal in the BL spectrum. After the subtraction, we expect that the only XP emission comprises the PL spectrum in Fig. R4 (left, blue curve). Indeed, the subtracted PL spectrum was identical to that of CDL that is mostly due to XP, confirming the validity of our method for isolating the PL spectrum of XP of BL from XM or XF emission. Therefore, the XP PL peak can be clearly distinguished from the XM peak in the D-A heterostructures. We now performed the plot of the dispersion, that is, the energy of XP vs. k_x/k_0 , using the subtracted PL spectra as the result is shown in Fig. R5, which showed the almost same degree of a strong energy-momentum dispersion as the dispersion seen when the raw PL spectra of BFP BL were used for the $E-k$ plot, indicating that partial inclusion of XM in PL spectra is not the main contribution to the observed angular dispersion of XP.

Fig. R4 Left: BFP PL spectra of m-MTDATA (black) and that of bilayer (red). Right: BFL spectrum (blue) of m-MTDATA/T2T bilayer after the subtraction of PL of m-MTDATA. BFL spectrum of CDL (magenta curve).

Fig. R5. Exciton energy vs relative momentum (k_x/k_0) of m-MTDATA/T2T bilayer before and after the subtraction of XM emission component.

As for the possible PL emission of T2T that the reviewer has pointed out, we agree that T2T have some PL emission in the exciplex range (Supplementary Fig. 2), despite the larger HOMO-LUMO gap than the energy of photons from the 405-nm laser. However, the PL of T2T upon 405-nm laser was negligible, as shown in Supplementary Fig. 2. The PL intensity ratio of T2T to m-MTDATA is lower than 1/1000 at the same measure condition ($t_{\text{ex}} = 10$ s and $P_{\text{in}} = 0.1 \mu\text{W}$), as shown in Fig. R6. Therefore, we can safely rule out the contribution of T2T emission to the PL spectra obtained with 405-nm laser excitation.

Fig. R6. **a** Supplementary Fig. 2. **b** PL intensity ratio between m-MTDATA and T2T

2. “The different angular dependence of the exciton and exciplex signals in Fig. 2 are not necessarily an indication of different optical dipole orientation. As explained in Ref. 41, they can both come from in-plane dipoles. As further shown in this Ref., a quantitative analysis of these BFP images is required to come to an unambiguous conclusion.”

Author reply: Thank you for giving us to clarify this point. We agree with the reviewer that the simple shape of the cross-sectional profile of the BFP PL image does not exclusively determine the orientation of the exciton dipoles. As the reviewer pointed out, the previous MoS₂/WSe₂ HS displayed a bright edge in the BFP image, even though its IX was concluded to be in-plane, as in Ref. 45 (Ref. 41 in our original submission). Therein, both in-plane (IP) and out-of-plane (OP) excitons display bright edges in the BFP PL images. However, we note that in Refs. 45 and 46, BFP image range extends to $k_x/k_0 \cong 1$ and the edge brightness of BFP is mostly due to the strong outcoupling of emitted photons at a so-called supercritical angle (SA) [new Ref. 48, James Shirley, F., et al. Supercritical Angle Fluorescence Characterization Using Spatially Resolved Fourier Plane Spectroscopy. *Anal. Chem.* **90**, 4263–4267 (2018)]. If BFP profile is limited within the SA, it is mostly determined by the intuitive picture given in Fig. 2a, where the IP and OP excitons show convex and concave profiles, respectively. This is the case for our BFP PL imaging in the original submission and in new Ref. 47 [Schuller, J. A., et al. Orientation of luminescent excitons in layered nanomaterials. *Nat. Nanotechnol.* **8**, 271-276 (2013)] too, where the profiles of the IP and OP excitons were convex and concave, respectively within $k_x/k_0 \leq 0.6$, as shown in Fig. R7.

Fig. R7. p- (bottom) polarized LDOS for IP (blue) and OP (red) dipoles. [new Ref. 47 Schuller et al., *Nat. Nanotechnol.*, **8**, 271-276 (2013)]

To provide the even more compelling experimental evidence, we performed the additional BFP imaging experiments by using a linear polarizer in front of the spectrometer and resolve the p- and s-polarized emissions on the BFP. The exciton dipole

moments vertical to the substrate should predominantly emit p-polarized (p-pol) light [new Ref. 47, Schuller, J. A., et al. Orientation of luminescent excitons in layered nanomaterials. *Nat. Nanotechnol.* **8**, 271-276 (2013)]. A linear polarizer was placed in front of the detector to observe the XP characteristics in the BL and CDL samples. Long-pass and short-pass optical filters were used to observe the target emissions (XP or XF), and the intensity profile at the BFP was observed using a CCD camera. In addition, we performed additional BFP experiments using differently prepared samples of the drop-cast film and the film prepared by the reprecipitation method (reprecipitated film) of blended m-MTDATA-T2T molecules, which were more disordered than the slowly deposited m-MTDATA/T2T bilayers. The BFP images of the pristine m-MTDATA was measured for the comparison purpose. As shown in Supplementary Fig. 8a and b, BL and CDL samples showed p-polarization-dominant emission, indicating the vertical direction (out-of-plane) of the dipole moments. In contrast, the drop-cast m-MTDATA/T2T film and the reprecipitated film that are more disordered samples showed no polarized emission pattern of XPs, as shown in Supplementary Fig. 8c and d because of the random orientation of the XP dipoles.

Through a series of additional experiments with the BL, CDL, drop-cast film, and reprecipitated film resolving s- and p-polarization, we confirmed that the characteristic concavity of the cross-sectional profile across the BFP occurs only for the XPs of the BL or CDL and doesn't occur for disordered sample systems (Supplementary Fig. 7). Note that the drop-cast film produces XPs at the same wavelength as the BL or CDL. Therefore, the systematic increase in the concavity of the BFP profile and angular chromism between the BL (or CDL) and drop-cast (or reprecipitated) films provides the strong evidence of the aligned dipoles of XP in the BL and CDL. These have been included in new sections 7 and 8 in the revised SI.

Supplementary Fig. 8. BFP PL images of XPs in **a** BL, **b** CDL, **c** drop-cast film, and **d** reprecipitated film, and of **e** XF in deposited m-MTDATA. White arrows indicate direction of linear polarization. XP emissions were filtered using 488-nm long-pass filter, and XF emission was filtered using 409-nm long-pass and 450-nm short-pass filters. It is noted that the measurement system of these BFP (for Supplementary Fig. 8**a-d**) is different from that for Fig. 2 and Supplementary providing slightly larger range of k_x .

Based on these arguments, a discussion has been added to the revised manuscript as the following: “In previous literature, the edge of BFP near at $k_x = \pm 1$ often displays intense PL regardless of the exciton orientation [Ref. 45, new Ref. 47], due to the highly efficient outcoupling near at supercritical angle [new Ref. 48]. However, the NA of the air objective lens used in our BFP PL imaging limits the range of k_x well within ± 1 and thus the scheme shown in Fig. 2a is generally valid, showing the characteristic convexity and concavity in BFP profile for in-plane or out-of-plane excitons, respectively [new Ref. 47].” (on page 10). “To understand the relationship between the XP E - k dispersion and the degree of dipole alignment of the XPs, additional experiments using more disordered films of donor m-MTDATA and acceptor T2T blending, where the orientation of XPs should be all random, were performed. The E vs. k relations of the XP peak from the BFP PL spectra and images of the blended drop-cast and reprecipitated films did not show the dispersive characteristics as shown in Supplementary Fig. 7 and Fig. 8. The negligibly small κ_{center} of drop-cast and reprecipitated films compared to those of bilayer and CDL by approximately an order of magnitude are listed in Supplementary Table 1. In addition, while XPs of drop-cast or reprecipitated films displayed the clear XP emission at the same wavelength as BL samples, the characteristic concavity of BFP cross-sectional profile was not observed in either of additionally prepared samples (Supplementary Fig. 7), confirming that observed concavity of BFP profile in BL samples originates from the vertically aligned nature of XP dipole orientation.” (on page 12)

3. “It is not true that a CT exciton is more stable than a molecular exciton, as claimed on p. 4. The opposite is the case because the exciton binding energy scales inversely with the distance of the e-h pair.”

Author reply: Thank you for the comment. As the reviewer pointed out, the exciton binding energy scales inversely with the distance between the electron and hole of the exciton. On page 4, we previously mentioned, “Therefore, more stable and long-lived XPs, such as interlayer and intermolecular CT excitons devices.”, By the “more stable XPs” in this sentence we implied that the Coulomb bound electron-hole pairs at heterojunction could hold their aligned dipole moment and energy for much longer owing to the long lifetime of XPs than XF, enabling the stable energy supply to the emissive dopants compared to fast-recombining intralayer excitons or intramolecular excitons (XF). To avoid confusion, we have modified the sentence as the following: “Therefore, the long-lived and aligned XP is a promising energy host in optoelectronic devices or a

platform for large interaction with other quasi particles like photon and phonon in organic photonic devices.”

4. “The notation $|D^*A\rangle$ etc. does not stand for the wavevector (as claimed on p. 8) but for the wave function of a certain state.”

Author reply: Thank you for the comment. The correct notation $|D^*A\rangle$ indicates a wavefunction. For quantum states, the word “wavevector” can be used interchangeably with the word “wavefunction.” In our study, the wavevector was used in the results of the BFP. Therefore, as the reviewer pointed out, we have corrected the notation of $|D^*A\rangle$ to “wavefunction” on page 8.

5. “And what is the reasoning for the parabolic dispersion relation of CT excitons on p. 11?”

Author reply: Thank you for the comment. In momentum (k) space, the energy of a quantum particle can be approximately described as a function of k_i ($i = x, y, z$), $E(k) = p^2/2m^* = \hbar^2 k^2/2m^*$. Here we regarded the CT excitons such as XPs as quantum quasi-particles with momentum $p = \hbar k$.

We have responded to all comments raised by the reviewers. The manuscript has been carefully revised based on the responses. Thank you for your comments.

Sincerely yours,
Jinsoo Joo
Professor
Department of Physics
Korea University
Seoul 02841
Republic of Korea
E-mail: jjoo@korea.ac.kr

Jeongyong Kim
Professor
Department of Energy Science

Sungkyunkwan University
Suwon 16419
Republic of Korea
E-mail: jkim@skku.edu

REVIEWERS' COMMENTS

Reviewer #1 (Remarks to the Author):

Thanks for providing additional data to the manuscript. Based on the small difference in device performance of the BL and CDL OLEDs (Supplementary Fig. 19d), I think it is not convincing enough to claim such effect is caused by dipole orientation of excitons, but simply experimental errors. Could the author include error bars for all the device characteristics measurements.

Reviewer #4 (Remarks to the Author):

The authors has carefully addressed the points raised by the referees in excellent manners. This new data, revised main text and supporting materials are satisfactory to convince this reviewer for recommending acceptance of this work for publication in this journal.

Reviewer #5 (Remarks to the Author):

Taking the function of referee #3 and checking the discussion and author's responses the following conclusions can be made:

1. The discussion concerning the terminology of whether an CT exciton is applicable to both 2D ordered structures and molecular disordered interfaces– to my opinion, CT exciton does not depend on the interface morphology but only on the extent of the interface polarity. Therefore, I support the author's viewpoint. At the same time, the CT exciton is not always equivalent to exciplex. Therefore, this term should be used with the great care. Usually, exciplex is formed due to excitation of one molecule, which leads to attraction of the other ground-state species, which then together emit light. I cannot see the evidence of that process in the manuscript. Therefore, I recommend the authors to limit usage of this term until they show evidence of exciplex behaviour in more detail.
2. I found all other author's responses to be satisfactory.

Overall, this is a nice work establishing a relation between CT excitons and orientation of interface dipoles which have a positive feedback on specific exciton properties.

Response Letter

NCOMMS-23-20378A

Title: Observation of aligned dipoles and angular chromism of exciplexes in organic molecular heterostructures

Authors: Sang-hun Lee, Taek Joon Kim, Eunji Lee, Dayeong Kwon, Jeongyong Kim, and Jinsoo Joo

Dear Reviewers:

We thank the reviewers for their valuable comments. Their comments have helped the improvement of the quality of the revised manuscript. Thank you for the acceptance of reviewer #4. To respond to the reviewer #1 comments and questions, we have included the error bars for the data of efficiencies of OLEDs (Supplementary Fig. 18, 19) and the discussion of the results considering the error range. To respond to the reviewer #5 comments, we have described the exciplex (XP) in more detail with the observed PL peak. Please see our responses, the revised manuscript, and Supplementary Information (SI) for more details.

Reviewer #1's comments

Comments: "Thanks for providing additional data to the manuscript. Based on the small difference in device performance of the BL and CDL OLEDs (Supplementary Fig. 19d), I think it is not convincing enough to claim such effect is caused by dipole orientation of excitons, but simply experimental errors. Could the author include error bars for all the device characteristics measurements."

Author reply: Thank you for the insightful comments. As the reviewer suggested, we have included the error bars for efficiencies of OLEDs in Supplementary Fig. 18 and 19.

New Supplementary Fig. 18 a I - V - L curves of BL (red) and CDL (black) OLEDs with error bars.

New Supplementary Fig. 19. a and d EQE, **b and e** CE, **c and f** PE of BL (red) and CDL (black) OLEDs as functions of bias and current density with error bars.

The following discussion based on the comments of the reviewer #1 (considering error bars) have been included in the Supplementary section 16:

“Supplementary Fig. 19 shows the EQE, current efficiency (CE), and power efficiency (PE) as functions of bias and current density for the BL (red) and CDL (black) OLEDs.

It is noted that the OLEDs for Supplementary Fig. 19 are new batch devices. The overall efficiency of the BL OLED was slightly higher than that of the CDL OLED in applied voltage dependence (Supplementary Fig. 19a–c). The EQE, CE, and PE of BL OLEDs showed comparable efficiencies with those of CDL OLEDs at high current density levels considering error bars (over 10 mA/cm², Supplementary Fig. 19d-f). In low current regime, the injected electron-hole recombination rate for radiative decay in the BL OLED is higher than that of the blended D-A molecules in the CDL OLED due to well defined D/A interface. The low EQE, CE, and PE of the BL and CDL OLEDs can be attributed to low outcoupling by the wide-angle emission of XP, power loss by hole leakage, and/or low charge balancing caused by the lack of functional layers such as the electron injection layer (EIL). A clear correlation between the EL peak positions with increasing bias and the dipole orientation of excitons/exciplexes has been discussed with Fig. 6c. The efficiencies of BL and CDL OLEDs (Supplementary Fig. 19d-f) at high current density levels did not show considerably differences (within error bars) related to the dipole orientation of exciton species.

Reviewer #4's comments

Comments: “The authors have carefully addressed the points raised by the referees in excellent manners. These new data, revised main text and supporting materials are satisfactory to convince this reviewer for recommending acceptance of this work for publication in this journal.

Author reply: Thank you for your acceptance.

Reviewer #5's comments

Comments: “Taking the function of referee #3 and checking the discussion and author's responses the following conclusions can be made:

1. The discussion concerning the terminology of whether an CT exciton is applicable to both 2D ordered structures and molecular disordered interfaces– to my opinion, CT exciton does not depend on the interface morphology but only on the extent of the

interface polarity. Therefore, I support the author's viewpoint. At the same time, the CT exciton is not always equivalent to exciplex. Therefore, this term should be used with the great care. Usually, exciplex is formed due to excitation of one molecule, which leads to attraction of the other ground-state species, which then together emit light. I cannot see the evidence of that process in the manuscript. Therefore, I recommend the authors to limit usage of this term until they show evidence of exciplex behaviour in more detail.

2. I found all other author's responses to be satisfactory.

Overall, this is a nice work establishing a relation between CT excitons and orientation of interface dipoles which have a positive feedback on specific exciton properties.”

Author reply: Thank you for the insightful comment. We agree with the reviewer's comment as “the CT exciton is not always equivalent to exciplex”. As pointed out by the reviewer, “exciplex is formed due to excitation of one molecule, which leads to attraction of the other ground-state species, which then together emit light.” In our study, we have used the exciplexes (XPs) in terms of charge transfer excitons between organic donor and acceptor molecules in organic heterojunctions (that is, hetero-intermolecular charge transfer excitons in π -conjugated organic molecules), which then together induce the light emission. These are depicted in Figure 1 c, d (or below Fig. R1 in this response letter). In our study, the PL peak due to the XP was observed at 575 nm (2.16 eV) at room temperature. The similar observations and the explanation of exciplex had been reported in references 5,13,18,26,27,28,29,30,etc.

Fig. R1 Left: Energy-band alignment with HOMO and LUMO levels of m-MTDATA and

T2T films and the formation of photoinduced CT exciton between m-MTDATA donor and T2T acceptor (i.e., XP). The HOMO and LUMO levels were estimated by using UPS and optical absorption spectra. Right: Normalized LCM PL spectra of m-MTDATA (black curve) and m-MTDATA/T2T bilayer (blue curve) thin films at 290 K. The PL peak due to the XP was observed at 2.16 eV.

For the clarification, we have added more explanation for the exciplex (XP) in the abstract, the main text, and the caption of Fig.1 as the following:

Abstract:

(before correction) The dipole characteristics of Frenkel excitons (XFs) and charge-transfer excitons (exciplexes; XPs) in organic heterostructures are important in organic photonics and optoelectronics.

(after correction) The dipole characteristics of Frenkel excitons (XFs) and charge-transfer excitons between donor and acceptor molecules in organic heterostructures such as exciplexes (XPs) are important in organic photonics and optoelectronics.

On page 4

(before correction) Quantum quasiparticles similar to the IXs of TMDCs and perovskites/QDs HSs can be achieved in π -conjugated organic systems as exciplexes (XPs), which are CT excitons formed in donor (D) and acceptor (A) heteromolecules^{5,26,27}.

(after correction) Quantum quasiparticles similar to the IXs of TMDCs and perovskites/QDs HSs can be achieved in π -conjugated organic systems as exciplexes (XPs). In this study, the XPs represent the CT excitons formed in organic donor (D) and acceptor (A) heteromolecules^{5,26,27}.

On page 7

(before correction) The LCM PL spectrum of the XM_{m-MT} was partially included. Supplementary Fig. 2 shows the m-MTDATA and T2T layers.

(after correction) The LCM PL spectrum of the XM_{m-MT} was partially included. It is noted that the observed XP_{m-MT/T2T} is due to the CT between m-MTDATA donor and T2T

acceptor. Supplementary Fig. 2 shows the m-MTDATA and T2T layers.

On page 8

(before correction) Therefore, the final quantum state of $[DA]^*$ can be described as $|[DA]^*\rangle = a_1|D^*A\rangle + a_3|D^+A^-\rangle$, as shown in Fig. 1d. The measured PL peak corresponding to XP is 575 nm (= 2.16 eV), as indicated in Fig. 1c, ...

(after correction) Therefore, the final quantum state of $[DA]^*$ can be described as $|[DA]^*\rangle = a_1|D^*A\rangle + a_3|D^+A^-\rangle$, as shown in Fig. 1d. In this study, $a_3|D^+A^-\rangle$ represents the quantum state of XP between m-MTDATA donor and T2T acceptor. The measured PL peak corresponding to XP is 575 nm (= 2.16 eV), as indicated in Fig. 1c, ...

The caption of Figure 1:

(before correction) **c** Normalized LCM PL spectra of m-MTDATA (black curve) and m-MTDATA/T2T bilayer (blue curve) thin films at 290 K. **d** Energy-band alignment with HOMO and LUMO levels of m-MTDATA and T2T films and the formation of photoinduced CT exciton (i.e., XP). The HOMO and LUMO levels were estimated by using UPS and optical absorption spectra.

(after correction) **c** Normalized LCM PL spectra of m-MTDATA (black curve) and m-MTDATA/T2T bilayer (blue curve) thin films at 290 K. The PL peak due to the XP was observed at 2.16 eV. **d** Energy-band alignment with HOMO and LUMO levels of m-MTDATA and T2T films and the formation of photoinduced CT exciton between m-MTDATA donor and T2T acceptor (i.e., XP). The HOMO and LUMO levels were estimated by using UPS and optical absorption spectra.

We have responded to all comments raised by the reviewers. The manuscript has been carefully revised based on the responses. Thank you for your comments.

Sincerely yours,

Jinsoo Joo

Professor

Department of Physics
Korea University
Seoul 02841
Republic of Korea
E-mail: jjoo@korea.ac.kr

Jeongyong Kim
Professor
Department of Energy Science
Sungkyunkwan University
Suwon 16419
Republic of Korea
E-mail: jkim@skku.edu